# Unify ML4TSP: Drawing Methodological Principles for TSP and Beyond from Streamlined Design Space of Learning and Search

**Yang Li**[12]**, Jiale Ma**[12]**, Wenzheng Pan**[1]**, Runzhong Wang**[3]**, Haoyu Geng**[1]**,
Nianzu Yang**[1]**, Junchi Yan**[12] *

[1] Sch. of Artificial Intelligence & Sch. of Computer Science, Shanghai Jiao Tong University
[2] Shanghai Innovation Institute    [3] Massachusetts Institute of Technology
https://github.com/Thinklab-SJTU/ML4TSPBench

## Abstract

Despite the rich works on machine learning (ML) for combinatorial optimization (CO), a unified, principled framework remains lacking. This study utilizes the Travelling Salesman Problem (TSP) as a major case study, with adaptations demonstrated for other CO problems, dissecting established mainstream learning-based solvers to outline a comprehensive design space. We present ML4TSPBench, which advances a unified modular streamline incorporating existing technologies in both learning and search for transparent ablation, aiming to reassess the role of learning and discern which parts of existing techniques are genuinely beneficial and which are not. This further leads to the investigation of desirable principles of learning designs and the exploration of concepts guiding method designs. We demonstrate the desirability of principles such as joint probability estimation, symmetry solution representation, and online optimization for learning-based designs. Leveraging the findings, we propose enhancements to existing methods to compensate for their missing attributes, thereby advancing performance and enriching the technique library. From a higher viewpoint, we also uncover a performance advantage in non-autoregressive and supervised paradigms compared to their counterparts. The strategic decoupling and organic recompositions yield a factory of new TSP solvers, where we investigate synergies across various method combinations and pinpoint the optimal design choices to create more powerful ML4TSP solvers, thereby facilitating and offering a reference for future research and engineering endeavors.

## 1 Introduction

Combinatorial Optimization (CO) has been a central challenge with its inherent complexity (e.g., NP-hardness). Recently, machine learning (ML) has been actively introduced to address CO problems, i.e., ML4CO (Bengio et al., 2021; Cappart et al., 2021), which brings practical advantages in both solving quality and speed especially when the instances are confined within a certain distribution (Bengio et al., 2021). Among CO problems, the Travelling Salesman Problem (TSP) has been representative as one of the most intensely studied problems in both ML4CO and Operations Research (OR) communities with one of the richest libraries of learning and search techniques (Hottung et al., 2021b; Fu et al., 2021; Min et al., 2023; Qiu et al., 2022; Hudson et al., 2022; Sun & Yang, 2023; Li et al., 2023b; Kool et al., 2018; Kwon et al., 2020; Kim et al., 2022). For solving CO problems, deep neural networks enjoy high parallel computing ability for speedup through one-shot inference. Meanwhile, ML could automatically uncover heuristics through data (Kool et al., 2018; Luo et al., 2023) for improvement against existing handcrafted rules, especially for some new problems or new instance distributions that call for experts to develop tailored solving schemes.

Recent advancements in learning-based solvers have significantly enhanced their solving performance and capacity to tackle larger-scale problems. These improvements have been propelled by a variety

*Correspondence author: yanjunchi@sjtu.edu.cn. The SJTU authors are partly supported by NSFC (92370201), Shanghai Municipal Science and Technology Major Project (2021SHZDZX0102).

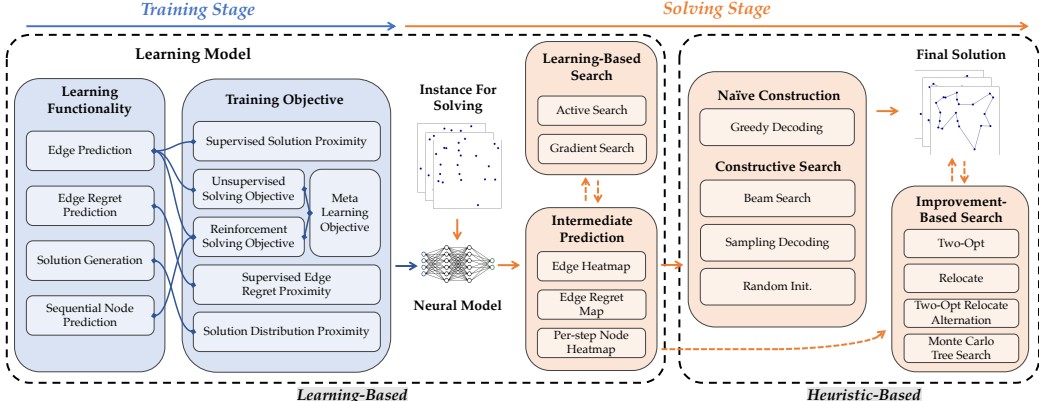

Figure 1: Overview of our proposed ML4TSPBench[1]. Blue and orange stand for training and testing processes, respectively. Dashed lines indicate the optional processes in pipelines.

of learning paradigms, encompassing supervised predictive modeling (Hottung et al., 2021b; Fu et al., 2021; Hudson et al., 2022), reinforcement learning (Qiu et al., 2022; Kool et al., 2018; Kwon et al., 2020; Kim et al., 2022), generative modeling (Sun & Yang, 2023; Li et al., 2023b), unsupervised learning (Min et al., 2023), meta learning (Qiu et al., 2022), etc. In modern approaches, learning and search elements are often interleaved with each other, and common designs among existing learning-based solvers may manifest in varying implementations, making their effects and contributions less transparent, let alone their intricate interplay. Meanwhile, the lack of crisp analysis of the whole ML4TSP system (a representative example for ML4CO) hinders the determination of the role of learning and the establishment of desirable principles for learning designs.

In this paper, we adopt a higher-level perspective to consider the whole system and advocate the necessity of inspecting and aligning various existing methods. We endeavor to decompose established mainstream learning-based solvers to outline a comprehensive design space and unify a modular streamline incorporating aligned learning and search technologies while minimizing disparities among methods, facilitating transparent ablation and systematic analysis. Fig. 1 presents the overview of our framework. Given instances and their (optional) labels e.g. reference solutions, the training stage involves different learning functionalities and training objectives to exploit the effective heuristics from data. Given instances, the solving stage follows a construction-refinement pipeline whereby a complete (feasible) solution is generated first then it is improved through the improvement-based search. The construction is based on the intermediate predictions produced by the trained model, and learning-based search is optionally applied to improve prediction quality by updating the parameters of the model or the intermediate predictions. Note the prediction results for auto-regressive methods using sequence models (Kool et al., 2018; Kwon et al., 2020) involve the interplay of neural models and construction methods to update the node heatmap prediction for different steps.

Based on this framework, we analyze the correlation between learning and solving objectives along with the efficacy of current ML4TSP endeavors within the outlined framework. This further leads to the summarization of the inherent desirable design principles among diverse learning methods and discerns key attributes that contribute to performance improvement. Specifically, minimized ablation studies are performed demonstrating the desirability of design principles such as joint probability estimation, symmetry solution representation, online optimization, and search friendliness. Additionally, we offer insights into how model predictions intervene with strong search methods like MCTS and highlight preferences in learning paradigms (e.g., supervised vs. unsupervised, autoregressive vs. nonautoregressive) regarding solving performance and scalability.

Leveraging our findings, we propose enhancements to existing methods to compensate for their missing attributes, thereby advancing performance and enriching the technique library. Examples of novel methods include node-level normalization for one-shot solution predictions (Joshi et al., 2019), symmetry introduction into the diffusion-based solvers (Sun & Yang, 2023; Li et al., 2023b) and the reinforcement solving objective of Qiu et al. (2022), the devised MCTS variants by decoupling the dependencies between solution initialization and local search within MCTS (Fu et al., 2021) and connecting its local search component with established constructive search methods, etc. Furthermore, we recom-

---

[1]The implementation is based on https://github.com/Thinklab-SJTU/ML4CO-Kit.

pose within the established technique library and propose a factory of solvers, among which the optimal ones achieve performance gains over previous SOTA. Beyond TSP, we also demonstrate adaptions of the framework to other CO problems and verify that the desirability of design principles discovered in ML4TSP evaluations still holds for other CO problems. Our implementation is carefully organized, featuring solver classes for various learning functionalities, facilitating easy development for further research and engineering practices. We aim for this work to streamline the development and evaluation process, offering a consistent implementation for verifying the effects of proposed methods efficiently.

## 2 PRELIMINARIES AND RELATED WORK

**Graph Combinatorial Optimization and Travelling Salesman Problem (TSP).** Following Karalias & Loukas (2020); Wang et al. (2022), we define $\mathcal{G}$ as the universe of CO instances by graphs $G(V, E) \in \mathcal{G}$, where $V$ and $E$ denote the node set and edge set respectively. Let $\mathbf{x} \in \{0, 1\}^N$ denote the optimization variable and $\Omega$ denotes the feasible set. A CO problem on $G$ aims to find a feasible $\mathbf{x} \in \Omega$ that minimize the given objective function $f(\cdot; G) : \Omega \to \mathbb{R}_{\geq 0}$. TSP is defined on an undirected complete graph $G = (V, E)$, where $V$ represents $n$ cities and each edge $E_{i,j}$ has a non-negative weight $\mathbf{D}_{i,j}$ representing the distance between cities $i$ and $j$. The problem is to find a Hamiltonian cycle of minimum weight in $G$. Here the dimension of the optimization variable $N = n^2$, and we denote the solution as a matrix $\mathbf{X} \in \{0, 1\}^{n \times n}$ where $\mathbf{X}_{i,j}$ indicates whether $E_{i,j}$ is included in $\mathbf{x}$. The objective function is: $f(\mathbf{X}; G) = \mathbf{D} \odot \mathbf{X} = \sum_{i,j} \mathbf{D}_{i,j} \cdot \mathbf{X}_{i,j}$.

**Related Works.** This is an early endeavor to rethink existing methodologies in ML4TSP while concurrently advancing the development of new techniques. It stands in contrast to Berto et al. (2023) which primarily focused on providing a unified implementation and evaluation of RL-based models without delving into deeper analyses of learning paradigms and proposing new methods. Joshi et al. (2020) tries to analyze from the generalization perspective, yet the investigated methods are confined to naive pipelines (Kool et al., 2018; Joshi et al., 2019) which is a very partial subset of the current ML4TSP system, and no general design principles or new methods are provided. Geng et al. (2023) provides a benchmark for the predictive combinatorial optimization. Pan et al. (2025) unifies a set of CO problems by reducing them into the general TSP form featured by distance matrices. Our framework tries to integrate the mainstream efforts of constructive ML4TSP solvers. The comprehensive nature of our framework facilitates a higher-level integration and analysis, thereby facilitating the development of a broader spectrum of research avenues within the community.

## 3 MODULAR FRAMEWORK FOR ML4TSP SOLVERS

### 3.1 DESIGN SPACE FOR TRAINING STAGE ALGORITHMS

The learning models generally receive input features from the instance graph, where the typical features optionally include node features indicating the 2D coordinates of the nodes, and edge features indicating the weight of the edges. The network backbones include GNN variants like Graph Convolutional Networks (GCN) (Kipf & Welling, 2016; Joshi et al., 2019), Graph Attention Networks (GAT) (Veličković et al., 2017; Kool et al., 2018), and Scattering Attention GNN (SAG) (Min et al., 2023). The algorithmic variances are reflected in the learning functionalities and training objectives presented below, where the correlations between these components are presented in Fig. 1.

#### 3.1.1 LEARNING FUNCTIONALITIES

Different learning functionalities are proposed to more efficiently utilize data for solving TSP, which involves what the neural models learn from pre-existing data.

**Edge Prediction.** Joshi et al. (2019); Fu et al. (2021); Qiu et al. (2022); Min et al. (2023) frame TSP solving as an edge prediction task, where each edge is classified as binary to determine whether it is in the solution. The prediction confidence of the edges (after softmax function) serves as a heatmap $\mathbf{H}$ denoting the probabilities of edges occurring in the tour, i.e., $\mathbf{H}_{i,j} = p(E_{i,j} \in \mathbf{S}^*)$ where $\mathbf{S}^*$ denotes the optimal solution represented in the matrix form.

**Edge Regret Prediction.** The global regret is defined as the cost of fixing a certain edge in the solution relative to the cost of a globally optimal solution (Hudson et al., 2022): $\mathbf{R}_{i,j} = \frac{f(\mathbf{S}^*_{i,j}; G)}{f(\mathbf{S}^*; G)} - 1$ where $\mathbf{R}_{i,j}$ denotes the regret of selecting $E_{i,j}$ in the solution, $\mathbf{S}^*_{i,j}$ denotes the optimal solution

with $E_{i,j}$ fixed, and $\mathbf{S}^*$ is the global optimal solution. This target contains more knowledge about how every edge benefits the optimization, especially for edges not included in the optimal solution, making it more conducive to searching. However, it necessitates significantly more computational resources to acquire the supervision.

**Solution Generation.** Deviating from the prediction of a single solution, generative modeling methods (Sun & Yang, 2023; Li et al., 2023b; Hottung et al., 2021a) endeavor to characterize a distribution of high-quality solutions for a given instance, i.e., estimating $p(\mathbf{S}|G)$. Solutions can be established by sampling from the distribution. This consideration of distributions makes predictions more diverse, thereby enhancing the search.

**Sequential Node Prediction.** Instead of global prediction in one shot, sequence models (Kool et al., 2018; Kwon et al., 2020; Kim et al., 2022) decompose it into n-step node predictions, where each step offers the prediction map of the next node selection based on the current state. The sequence models generally follow an attention based encoder-decoder model where the encoder produces embeddings of all input nodes and the decoder produces the per-step node heatmap predictions $\mathbf{h}_j^{(i)} = p(\pi_i = j | \pi_{<i})$ where $\mathbf{h}_j^{(i)}$ is the $j$-th element of step $i$'s prediction map and $\pi$ is the permutation representation of the solution.

### 3.1.2 Training Objectives

Training objectives are proposed to achieve the learning functionalities in Sec. 3.1.1. This subsection discusses supervised and unsupervised methods sequentially. Supervised methods effectively leverage the readily available solved data, while unsupervised methods bypass the need to acquire supervision.

**Supervised Solution Proximity.** This objective encourages the model's prediction, e.g. heatmap $\hat{\mathbf{H}} \in [0,1]^{n \times n}$, to approximate the reference solutions obtained by an alternative solver. Based on the binary classification for edges, the (weighted) binary cross-entropy loss is widely adopted:

$$\mathcal{L} = -\sum_{i,j} w_1 \cdot \mathbf{S}_{i,j} \cdot \log(\hat{\mathbf{H}}_{i,j}) + w_0 \cdot (1 - \mathbf{S}_{i,j}) \cdot \log(1 - \hat{\mathbf{H}}_{i,j}) \tag{1}$$

where $w_0, w_1$ are class weights since the classification task is unbalanced towards the negative class.

**Supervised Edge Regret Proximity.** This objective encourages the model to predict reliable edge regrets $\hat{\mathbf{R}} \in \mathbb{R}^{n \times n}$ to approximate the reference regrets $\mathbf{R}$, by minimizing $\mathcal{L} = ||\hat{\mathbf{R}} - \mathbf{R}||_2$.

**Solution Distribution Proximity.** Prominent models for the generative objective encompass diffusion models (Sun & Yang, 2023; Li et al., 2023b) and variational autoencoders (VAE) (Hottung et al., 2021a), to maximize the conditional likelihood estimation $\mathbb{E}[\log p_\theta(\mathbf{S}|G)]$, where $\theta$ is the model parameters. The models are typically optimized through the evidence lower bound (ELBO), where $q$ is the posterior and $\mathbf{Z}$ is the latent variable.

$$\mathcal{L} = -\mathbb{E}_{q(\mathbf{Z}|\mathbf{S},G)}\left[\log \frac{p_\theta(\mathbf{S}, \mathbf{Z}|G)}{q(\mathbf{Z}|\mathbf{S}, G)}\right] \geq \mathbb{E}\left[-\log p_\theta(\mathbf{S}|G)\right] \tag{2}$$

Unsupervised methods typically optimize using the solving objective. However, the non-differentiable mapping from continuous, unconstrained space to discrete, highly constrained space suffers gradient truncation. Thus, unsupervised loss based on constraint penalty and reinforcement solving loss are employed, and meta-learning can be further adapted to enhance generalization.

**Unsupervised Solving Objective.** Min et al. (2023) proposes penalties over the Hamiltonian cycle and no self-loop constraints to regularize the predicted assignment matrix $\hat{\mathbf{T}}$ and heatmap $\hat{\mathbf{H}}$. $\hat{\mathbf{T}}$ is obtained by enforcing column-wise softmax to the network output, where $\hat{\mathbf{T}}_{\cdot,j} \in [0,1]^{n \times 1}$ indicates the next-node probabilities at $i$ step, and $\hat{\mathbf{H}}$ is calculated through $\hat{\mathbf{H}} = \hat{\mathbf{T}}_{\cdot,n}\hat{\mathbf{T}}_{\cdot,1}^\top + \sum_{t=1}^n \hat{\mathbf{T}}_{\cdot,t}\hat{\mathbf{T}}_{\cdot,t+1}^\top$:

$$\mathcal{L} = \sum_{i,j} \hat{\mathbf{H}}_{i,j}\mathbf{D}_{i,j} + \lambda_1 \sum_i (\sum_j \hat{\mathbf{T}}_{i,j} - 1)^2 + \lambda_2 \sum_i \hat{\mathbf{H}}_{i,i}. \tag{3}$$

The second and third term denotes the Hamilton cycle and no self-loop constraint, respectively.

**Reinforcement Solving Objective.** With the predicted probability distribution $p_\theta(\mathbf{S}|G)$, we can sample solution tours. This objective optimizes the expectation of the tour length by gradient estimated by REINFORCE (Williams, 1992) with baseline $b(G)$:

$$\mathcal{L} = \mathbb{E}_{p_\theta(\mathbf{S}|G)}\left[f(\mathbf{S}; G)\right] \quad \text{and} \quad \nabla\mathcal{L} = \mathbb{E}_{p_\theta(\mathbf{S}|G)}\left[(f(\mathbf{S}; G) - b(G))\nabla \log p_\theta(\mathbf{S}|G)\right] \tag{4}$$

where the baseline function independently estimates the expected cost to reduce the variance of the gradients. The sampling construction method presented in Sec. 3.2.2 enables sampling solutions from the predicted edge heatmap and per-step node heatmaps to achieve $p_\theta(\mathbf{S}|G)$.

**Meta-Learning Objective.** This objective (Finn et al., 2017; Qiu et al., 2022) aims to train models to acquire meta-knowledge from diverse problem instances, improving generalization when faced with previously unseen instances. In CO, model parameters and model predictions can be further updated during testing through a learning-based search in Sec. 3.2.1. The objective tries to better predict a model initialization or prediction that can achieve the best solving results after the learning-based search in the solving phase. Suppose the edge heatmap $\hat{\mathbf{H}}$ is optimized to $\hat{\mathbf{H}}'$, then the meta-learning objective is $\mathcal{L} = \mathbb{E}_{p_s(\mathbf{S}|G,\hat{\mathbf{H}}')}[f(\mathbf{S};G)]$ where $s(\cdot)$ denotes the sampling decoding function.

## 3.2 DESIGN SPACE FOR SOLVING STAGE ALGORITHMS

The solving stage does not involve any pre-existing data but tries to optimize the solutions or the predictions over unseen instances received on the fly, as shown in Fig. 1.

### 3.2.1 LEARNING-BASED SEARCH

Motivated by the divergence between the training stage, which optimizes average performance across historical instances; and the solving stage that optimizes performance for each new instance, the work Hottung et al. (2021b); Qiu et al. (2022); Li et al. (2023b) design learning-based search in the solving stage to perform tailored optimization for testing instances, which improves prediction by updating the model parameters or the intermediate predictions.

**Active Search.** With objectives that do not require labels, active search (Hottung et al., 2021b; Qiu et al., 2022) further fine-tunes (partial) network parameters or intermediate model predictions, e.g. heatmaps, over each testing instance with solving objective updates of Eq. 4 using gradient decent.

**Gradient Search.** Based on diffusion modeling (Sun & Yang, 2023), T2T (Li et al., 2023b) introduces objective guidance into the denoising process of diffusion, adjusting the denoising to an objective-minimizing direction and rewriting the original learned denoising function $p_\theta(\mathbf{S}_{t-1}|\mathbf{S}_t, G)$ to

$$p_\theta(\mathbf{S}_{t-1}|\mathbf{S}_t, G, y^*) = Zp_\theta(\mathbf{S}_{t-1}|\mathbf{S}_t, G)\exp([-\nabla_{\mathbf{S}_t}\tilde{l}(\mathbf{S}_t; G)]^\top \mathbf{S}_{t-1}) \tag{5}$$

where $y^*$ is the optimal objective score given the instance $G$, $\tilde{l}$ denotes the objective estimation function. This technique allows for further refinement of the predicted heatmaps by iteratively adding noise to the current solution and then denoising it to an expected better one.

### 3.2.2 CONSTRUCTION METHODS

Construction methods typically leverage the predictions and the graphs to decode a complete solution.

**Greedy Decoding.** As a naive construction method, it sequentially inserts edges/nodes with the highest confidence in the heatmap to the partial solution for a feasible solution.

**Sampling Decoding.** It involves sampling edges or nodes based on their confidence scores in the heatmaps. Denoting $\pi$ as the permutation representation of $\mathbf{S}$, based on the edge heatmap $\hat{\mathbf{H}}$, the $t$-step node heatmap prediction can also be estimated from the edge heatmap $\hat{\mathbf{h}}^{(t)} = \frac{\exp(\hat{\mathbf{H}}_{\pi_{t-1},\pi_t})}{\sum_{j=i}^n \exp(\hat{\mathbf{H}}_{\pi_{t-1},\pi_t})}$.

**Beam Search.** At each step, beam search (Joshi et al., 2019) explores the heatmap by expanding the $b$ most probable edge connections among the nodes neighbors and saving the best $b$ partial solutions for the next step. The final prediction is the tour with the highest probability among the $b$ complete tours.

**Random Initialization.** For strong improvement-based search methods like MCTS, random initialization of tours can also produce plausible results.

### 3.2.3 IMPROVEMENT-BASED SEARCH

On top of the constructed complete solutions, improvement-based search further iteratively refines solutions by exploring within a limited neighborhood of the current solution.

**Two-Opt and Relocate.** Two-Opt swaps two nodes in the tour, dividing it into three segments, then reassembles the tour by reversing the middle segment. The relocate operator moves a single node

within the tour. The operators can be performed on the original solution to find lower-cost ones. These two operators can be combined by alternating between using either operator until no better solution can be found, which we call local search (LS) following Hudson et al. (2022) in the rest of the paper.

**Monte Carlo Tree Search.** The MCTS solver (Fu et al., 2021) iteratively proposes initial solutions and improves them to enhance solution quality. In each iteration, this process includes three components: 1) state initialization constructs a solution based on a given heatmap; 2) Two-Opt improvement explores a small neighborhood by Two-Opt until no improvements can be found; 3) MCTS based on K-Opt iteratively samples K-Opt operators based on the heatmap to improve the solution and update the heatmap for subsequent iterations. Typically, thousands of iterations of these steps are involved in an MCTS solver. However, we observed that increasing the number of outer iterations results in diminishing marginal improvements in heatmap updating. Thus, we reduce the number of outer iterations to one round, separating the construction phase and the improvement-based search phase. We adopt the latter as a design choice for improvement-based search, which can be integrated with construction methods such as greedy, sampling, beam search, and random initialization to generate MCTS variants.

**Prediction Guidance.** Improvement-based search can be guided by edge regret prediction (Hudson et al., 2022), where the edge scores for penalizing is calculated as $\frac{\hat{\mathbf{R}}_{i,j}}{1+n_{i,j}}$, where $n_{i,j}$ is the number of penalties assigned to $E_{i,j}$. The edges of maximum utility are penalized and the search is applied only to the penalized edge to maximize the chance of finding lower-cost solutions.

## 4 EMPIRICAL EVALUATION

This section tries to answer the following questions and proposes new techniques: **Q1.** *How does learning benefit testing stage problem solving?* **Q2.** *What are the desirable design principles for ML4TSP methods?* **Q3.** *Which combination can enhance solving performance?* The major empirical evaluation is conducted on TSP, yet Appendix C also shows examples of adapting the ML4TSP framework to other problems like Capacitated Vehicle Routing Problem (CVRP) and Maximum Independent Set (MIS) and verify whether conclusions drawn from TSP still hold for other problems.

### 4.1 EXPERIMENTAL SETUP

**Datasets.** A TSP instance includes $N$ 2-D coordinates and a reference solution. Instances are generated via uniformly sampling $N$ nodes from the unit square $[0, 1]^2$. Main experiments are performed on TSP on scales including 50, 100, and 500. We also include TSP-1000 and the real-world TSPLIB[1] dataset for generalization evaluation. The reference solutions for all problem scales are labeled by LKH-3 (Helsgaun, 2017). The test set for TSP-50/100 is taken from Kool et al. (2018) with 1,280 instances, and the test set for TSP-500/1000 is from Fu et al. (2021) with 128 instances.

**Metrics.** 1) Drop: the relative performance drop w.r.t. length compared to the optimal solutions (obtained by the exact solver Concorde (Applegate et al., 2006) in this paper); 2) Time: the average computational time per instance; 3) Length: the average total distance of the solved tours w.r.t. the corresponding instances i.e. the objective. Note the main paper omits this metric and indicates solution quality primarily by drop for clearness. Full results are presented in Appendix Table 9.

**Method Indicator.** We explore various combinations within the design space depicted in Fig. 1, making it impractical to label each variant individually. Thus, we directly list the design choices of the evaluated methods in the tables as indicators. Note the ablation may lead to convergence towards a subset of methods with common designs. In such cases, shared design choices are omitted from the table and explained in the caption for clarity. **To correspond to prior methods, please refer to Table 9.** To simplify method indication in experiments, we adopt the following abbreviations for certain design choices in our ML4TSP framework: 1) Learning Functionalities: Edge Prediction → Edge Pred, Edge Regret Prediction → Regret Pred, Solution Generation → Generation, Sequential Node Prediction → Sequential; 2) Training Objectives: Supervised Solution Proximity → SL, Supervised Edge Regret Proximity → SL Regret, Unsupervised Solving Objective → UL, *Reinforcement Solving Objective* → RL, Solution Distribution Proximity → Generative.

---

[1]http://comopt.ifi.uni-heidelberg.de/software/TSPLIB95/

Table 1: Joint probability estimation ablation based on methods with *Edge Pred* as the learning functionality. $8\times$ Greedy: sample 8 heatmaps in parallel then enforce greedy.

| Objective | Joint Prob. | Solving Stage | | TSP-50 | | TSP-100 | | TSP-500 | |
| --- | --- | --- | --- | --- | --- | --- | --- | --- | --- |
| | | Construction | Impr. Search | Drop↓ | Time | Drop↓ | Time | Drop↓ | Time |
| SL | ✗ | Greedy | – | 1.530% | 7.6ms | 7.078% | 10.6ms | 12.237% | 0.1s |
| SL + Node Norm | ✔ | Greedy | – | 0.674% | 7.4ms | 2.721% | 10.1ms | 8.496% | 0.1s |
| Generative | ✔ | Greedy | – | **0.076%** | 0.229s | **0.186%** | 0.4s | **6.695%** | 1.6s |
| SL | ✗ | Greedy | 2-Opt | 0.115% | 7.6ms | 0.649% | 12.0ms | 2.089% | 0.1s |
| SL + Node Norm | ✔ | Greedy | 2-Opt | 0.082% | 7.4ms | 0.292% | 11.7ms | 1.325% | 0.1s |
| Generative | ✔ | Greedy | 2-Opt | **0.046%** | 0.2s | **0.077%** | 0.4s | **0.883%** | 1.6s |
| SL | ✗ | Beam (n=1280) | – | 0.023% | 1.1s | 0.673% | 2.3s | 23.629% | 11.6s |
| SL + Node Norm | ✔ | Beam (n=1280) | – | 0.016% | 1.1s | 0.545% | 2.2s | 21.678% | 13.7s |
| Generative | ✔ | $8\times$ Greedy | – | **0.007%** | 0.7s | **0.011%** | 2.5s | **1.443%** | 10.3s |
| SL | ✗ | Beam (n=1280) | 2-Opt | 0.015% | 1.1s | 0.149% | 2.3s | 2.431% | 12.9s |
| SL + Node Norm | ✔ | Beam (n=1280) | 2-Opt | 0.012% | 1.1s | 0.156% | 2.3s | 2.189% | 14.0s |
| Generative | ✔ | $8\times$ Greedy | 2-Opt | **0.006%** | 0.7s | **0.009%** | 2.5s | **0.320%** | 10.4s |

## 4.2 LEARNING GENERALLY IMPROVES SOLVING WITH COMPATIBLE SEARCH ALGORITHMS

This subsection endeavors to answer **Q1**. Apart from the training objectives that optimize the solving performance, the reference proximity objectives are not directly related to the solving target. Since the solving performance relies on the synergy of neural predictions and search strategies, the correlation between the estimation accuracy in training and the final solving performance remains unclear. To verify this, Fig. 2 provides the scatter plots for the correlation between solving performance and the learning loss across reference proximity objectives on TSP-100 test sets.

Overall, there exists a positive correlation between learning quality and problem-solving performance. However, the introduction of a more robust search mechanism tends to mitigate the advantages of neural predictions. Moreover, once the learning loss surpasses a certain threshold (e.g., less than 0.003 for the distribution ob-

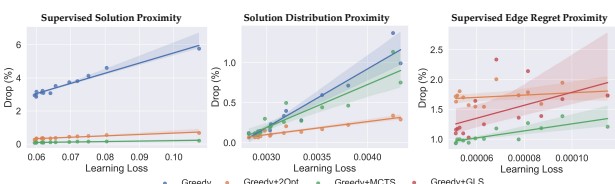

Figure 2: Performance correlation of three reference proximity objectives. GLS: guided local search.

jective), the efficacy of translating improvements in neural predictions into enhancements in problem-solving quality may diminish. This suggests a potential bottleneck in the capacity of the learning phase to enhance the testing phase. Regarding the prediction of edge regret, we find that the accuracy of regret estimation minimally impacts improvement-based search methods without heatmap guidance, such as Two-Opt. More results of the effects of learning to MCTS are presented in Appendix E.8.

## 4.3 DESIRABLE DESIGN PRINCIPLES THAT BENEFITS ML4TSP SOLVING

This subsection delves into crucial design principles in learning algorithms to answer **Q2**. We generalize overarching design principles with wide applicability across various learning designs, serving as foundational guidance that inspires methodological proposals. Building upon these principles, we introduce novel designs seamlessly integrated into existing pipelines to boost solving performance and validate the effectiveness of adhering to these principles throughout the process.

### 4.3.1 JOINT PROBABILITY ESTIMATION HELPS CAPTURE VARIABLE CORRELATIONS

This subsection validates the importance of joint probability estimation over solution variables in loss design. In the learning functionality of edge prediction, the *SL* training objective (Joshi et al., 2019; Fu et al., 2021) generally enforces supervision over each edge's binary classification results. This is based on the assumption of mutual statistical independence of the individual solution parameters, such that the solution can be represented by a product of Bernoulli distributions for the individual solution variables Sanokowski et al. (2023). However, since the CO solutions are highly constrained with strong correlations among the solution variables, such an assumption can lead to a performance drop.

With GNN raw predictions $\mathbf{F} \in \mathbb{R}^{n \times n \times 2}$, which correspond to $n \times n$ edges, prior methods utilize the softmax operation over the last dimension to ensure that the probabilities of positive and negative edges sum up to 1. The cross-entropy loss in Eq. 1 involves $n \times n$ edges' binary classification, lacking a general capture of joint probability over the solution variables. To this end, we propose node-wise probability normalization, which normalizes the positive edge probabilities associated with each

Table 2: Symmetry ablation based on methods with *Generative* as training objective.

| Symmetry | Solving Stage | | TSP-500 | |
|---|---|---|---|---|
| | Construction | Impr. Search | Drop↓ | Time |
| ✗ | Greedy | – | 9.639% | 1.2s |
| ✔ | Greedy | – | **6.695%** | 1.6s |
| ✗ | Greedy | 2-Opt | 1.524% | 1.2s |
| ✔ | Greedy | 2-Opt | **0.883%** | 1.6s |
| ✗ | 8× Greedy | – | 4.613% | 6.9s |
| ✔ | 8× Greedy | – | **1.443%** | 10.3s |
| ✗ | 8× Greedy | 2-Opt | 0.686% | 6.9s |
| ✔ | 8× Greedy | 2-Opt | **0.320%** | 10.4s |

Table 3: Symmetry ablation based on methods with *RL* as the training objective. AS: active search.

| Functionality | Symmetry | Solving | TSP-50 | | TSP-100 | |
|---|---|---|---|---|---|---|
| | | | Drop↓ | Time | Drop↓ | Time |
| Edge Pred + Rand. Start | ✗ | Sampling | 5.238% | 11.4ms | 9.364% | 94.5ms |
| | ✔ | Sampling | **2.157%** | 72.7ms | **4.885%** | 0.1s |
| Edge Pred + Rand. Start | ✗ | AS+Sampling | 3.086% | 3.2s | 4.234% | 6.7s |
| | ✔ | AS+Sampling | **0.823%** | 4.9s | **2.089%** | 9.3s |
| Sequential + MultiStart + Sym. Baseline | ✗ | Greedy | 1.040% | 0.1ms | 2.520% | 0.2ms |
| | ✔ | Greedy | **0.179%** | 1.7ms | **1.641%** | 1.6ms |
| | ✔ | Greedy | 0.877% | 0.6ms | 2.209% | 0.6ms |
| Sequential + MultiStart + Sym. Baseline | ✗ | Sampling | 1.212% | 0.1ms | 2.846% | 0.2ms |
| | ✔ | Sampling | **0.156%** | 1.3ms | **1.618%** | 1.6ms |
| | ✔ | Sampling | 1.041% | 0.6ms | 2.593% | 0.6ms |

Table 4: Ablation on online optimization property, which can be achieved by learning-based search.

| Functionality | Objective | Solving Stage | | | TSP-50 | | TSP-100 | | TSP-500 | |
|---|---|---|---|---|---|---|---|---|---|---|
| | | Learning Search | Construction | Impr. Search | Drop↓ | Time | Drop↓ | Time | Drop↓ | Time |
| Generation | Generative | ✗ | Greedy | – | 0.076% | 0.2s | 0.186% | 0.4s | 6.734% | 1.6s |
| | Generative | Gradient Search | Greedy | – | **0.015%** | 0.7s | **0.023%** | 0.8s | **2.675%** | 3.7s |
| | Generative | ✗ | Greedy | 2-Opt | 0.046% | 0.2s | 0.077% | 0.4s | 0.811% | 1.6s |
| | Generative | Gradient Search | Greedy | 2-Opt | **0.012%** | 0.7s | **0.023%** | 0.8s | **0.445%** | 3.8s |
| Edge Pred | Meta + RL | ✗ | Sampling | – | 4.310% | 35.9ms | 4.885% | 0.1s | 13.046% | 1.3s |
| | Meta + RL | Active Search | Sampling | – | **3.006%** | 2.9s | **3.940%** | 8.5s | **6.924%** | 274.8s |
| | Meta + RL | ✗ | Sampling | 2-Opt | 1.821% | 0.1s | 2.880% | 0.2s | 6.228% | 6.4s |
| | Meta + RL | Active Search | Sampling | 2-Opt | **1.549%** | 4.4s | **2.176%** | 8.9s | **4.322%** | 277.0s |

node sum up to 2 and the negative probabilities sum to $n-2$: $\mathbf{F}'_{i,j} = \frac{\exp(\mathbf{F}_{i,j})}{\sum_{k=i}^{n}\exp(\mathbf{F}_{i,k})} \cdot \begin{bmatrix} n-2 & 0 \\ 0 & 2 \end{bmatrix}$.
Then it clips the probability to (0,1). The predicted heatmap is obtained by $\hat{\mathbf{H}} = \mathbf{F}'[:,:,1]$.

Under the edge prediction functionality, we also include methods of solution distribution proximity objective into comparison, which considers a solution as the whole entity, allowing it to naturally adhere to the joint probability distribution. Table 1 shows the ablation. Methods with joint probability estimation generally achieve lower drops, while relying solely on the proposed node-wise probability normalization can bring about significant performance improvement without any additional computational overhead. The generative objective can achieve much lower drops, albeit with the added computational cost of multistep inference in the diffusion model.

### 4.3.2 SYMMETRY SOLUTION REPRESENTATION INTRODUCES EQUIVALENCE AWARENESS

A tour solution holds symmetry or permutation invariance (Kwon et al., 2020). For instance, let tour $\pi = \{v_1, v_2, v_3, v_4\}$ be the optimal solution of some given 4-node TSP instance. Then tours with different starting points, e.g. $\pi' = \{v_2, v_3, v_4, v_1\}$, and tours in a counterclockwise direction, e.g. $\pi'' = \{v_4, v_3, v_2, v_1\}$, all represent the same optimal solution. Since CO aims to produce optimal solutions, the neural model is expected to be aware of the equivalences among such sequences and should be optimized to consistently yield the same solution regardless of the variances in forms. This subsection endeavors to introduce the symmetry solution representation attribute to methods lacking symmetry consideration within the framework, thereby enhancing problem-solving performance.

For *Generation* learning functionality with *Generative* training objective, Sun & Yang (2023); Li et al. (2023b) aim to generate directed graph representations (heatmaps) of solutions with directed graph supervision enforced in the optimization of Eq. 2. We replace the directed graph representation for learning with the undirected one, which naturally accommodates the symmetry, and discover a notable performance gain in Table 2. This does not explicitly increase computation during the learning process, but it introduces a slight overhead during solution decoding.

For the *RL* objective, methods decode solutions sequentially during optimization. For instance, Qiu et al. (2022) sequentially sample multiple solutions based on the predicted edge heatmap, all starting from the same point, to estimate the route length expectations and the REINFORCE gradients, and sequence models (Kool et al., 2018) predict node sequence using per-step node heatmaps for optimization. To achieve symmetry, we propose to enforce a random starting point decoding policy in REINFORCE updates (in training and active search) for methods with *Edge Pred* functionality, and refer to Kwon et al. (2020) to optimize over diverse rollouts with multiple starting points as well as use symmetry baselines (Kim et al., 2022) for *Sequential* functionality. These techniques raise the model's awareness of symmetry. Table 3 presents solving results on TSP-50 and 100, showing a notable performance gain by introducing symmetry, yet with additional computational overhead.

### 4.3.3 ONLINE OPTIMIZATION COMPLEMENTS GENERALIZATION CAPABILITY

In order to bridge the gap between training for average performance and solving for performance in new instances, instance-specific optimization can be helpful. We verify the effectiveness of techniques that optimize in the testing phase in Sec. 3.2.1: the active search for methods with *Edge Pred* as the functionality and the gradient search for methods with *Generation* as the functionality. Table 4 ablates learning-based search across different baselines, indicating a promising gain with learning-based search, however at the cost of additional computation.

### 4.3.4 SEARCH FRIENDLINESS CONTRIBUTES TO LEARNING-SEARCH SYNERGY

Techniques that consider how to better integrate learning with search can benefit problem-solving. We verify the effectiveness of search friendliness designs including enhancing the informativeness of prediction information (e.g. predicting edge regrets), promoting diversity in the predicted solutions (e.g. modeling solution distribution), and incorporating the search

Table 5: Ablation on search friendliness property.

| Functionality | Objective | Search Frnd. | Solving Stage | TSP-50 | | TSP-100 | |
|---|---|---|---|---|---|---|---|
| | | | | Drop↓ | Time | Drop↓ | Time |
| Edge Pred | SL | ✗ | Guided LS (t=1s) | 0.614% | 1.1s | 3.319% | 1.2s |
| Regret Pred | SL Regret | ✔ | Guided LS (t=1s) | **0.098%** | 1.1s | **1.049%** | 1.4s |
| Edge Pred | SL | ✗ | Random + MCTS | 0.002% | 62.5ms | 0.001% | 0.4s |
| Regret Pred | SL Regret | ✔ | Random + MCTS | **0.000%** | 0.2s | 0.179% | 1.8s |
| Sequential | RL | ✗ | Sampling | 5.238% | 11.4ms | 9.364% | 94.5ms |
| Sequential | Meta + RL | ✔ | Sampling | **2.157%** | 72.7ms | **4.885%** | 0.1s |
| Sequential | RL | ✗ | AS+Sampling | 3.086% | 3.2s | 4.234% | 6.7s |
| Sequential | Meta + RL | ✔ | AS+Sampling | **0.823%** | 4.9s | **2.089%** | 9.3s |

process into optimization (e.g. through meta learning to learn optimal initialization for solving stage). Table 5 verifies the solving performance of providing more informative prediction and adopting meta learning, which may also introduce slight overhead. We notice that for MCTS, the performance gain of predicting edge regrets is minor, which may stem from the compatibility between the techniques. For the effectiveness of prediction diversity, Table 2 demonstrates a significant improvement in solving performance achieved by employing multiple sampling, which is contributed by increased diversity.

### 4.4 STREAMLINING THE ML4TSP DESIGN SPACE DISCOVERS MORE POWERFUL SOLVERS

This section investigates **Q2&3** by exploring potential recomposition among separable components in our framework, with the full results presented in Appendix Table 9.

**Evaluation of Heuristic Search.** We categorize search strategies outside learning as a construction-improvement paradigm, where constructive search and improvement-based search are orthogonal. Table 9 shows that the most proper heuristic search strategy varies for the specific learning functionalities and objectives. Interestingly, even

Table 6: Heuristic search ablation based on *Edge Pred* functionality and *SL* training objective.

| Solving Stage | | TSP-50 | | TSP-100 | | TSP-500 | |
|---|---|---|---|---|---|---|---|
| Construction | Impr. Search | Drop↓ | Time | Drop↓ | Time | Drop↓ | Time |
| Greedy | – | 1.530% | 7.6ms | 7.078% | 10.6ms | 12.237% | 0.1s |
| Beam (n=1280) | – | 0.023% | 1.1s | 0.673% | 2.3s | 23.629% | 11.6s |
| MCTS Solver (10t) | | 0.006% | 56.7ms | 0.006% | 0.217s | 0.433% | 10.1s |
| Greedy | Two-Opt | 0.115% | 7.6ms | 0.649% | 12.0ms | 2.089% | 0.1s |
| Greedy | Guided LS (t=1s) | 0.614% | 1.1s | 3.319% | 1.2s | 7.682% | 45.5s |
| Greedy | MCTS | 0.018% | 8.4ms | 0.154% | 16.8ms | 0.838% | 1.3s |
| Beam (n=1280) | Two-Opt | 0.015% | 1.1s | 0.149% | 2.3s | 2.431% | 12.9s |
| Beam (n=50) | MCTS | 0.005% | 0.1s | 0.009% | 0.5s | **0.235%** | 56.3s |
| Sampling (n=50) | MCTS | 0.002% | 0.1s | 0.001% | 0.5s | 0.396% | 58.1s |
| Random (n=10) | MCTS | 0.004% | 17.5ms | 0.011% | 81.4ms | 0.519% | 11.2s |
| Random (n=50) | MCTS | **0.002%** | 62.5ms | **0.001%** | 0.4s | 0.301% | 55.2s |

the more intricate MCTS method does not necessarily yield the best performance. In cases where the output heatmap closely approximates feasible solutions and exhibits high discreteness (e.g. the *Generative* objective), the effectiveness of utilizing the heatmap to guide the improvement search often falls short when compared to methods with smoother and more continuous heatmaps (e.g. *SL* and meta *RL* objectives for *Edge Pred*). Due to space limitation, Table 6 adopts the fundamental solution proximity objective for edge prediction as an example to illustrate the effectiveness and recomposing performance of heuristic search methods implemented in our framework. The decoupled improvement MCTS can be combined with other construction methods to form variants like *Greedy_MCTS*, *Sampling_MCTS*, *Beam_MCTS*, and *Random_MCTS*, which utilize different constructive search for MCTS initialization, showing improvement in both solving performance and efficiency compared to the original MCTS solver (Fu et al., 2021). We use different time slices for varying problem scales, with the time for a single time slice denoted as $t$. We control the search process of MCTS by adjusting the number of time slices. More details are given in Appendix F.3.

**Recomposing to the Optimal.** We recompose the learning and search techniques in our framework to produce a factory of solvers. Table 7 presents the best three method combinations for every learning functionality. The recomposed solvers achieve SOTA performance that reaches 0.000% for TSP-50, 0.001% for TSP-100, and 0.142% for TSP-500. We find that top-performing methods remain closely tied to robust search such as MCTS, while generative models, specifically diffusion

Table 7: Optimal recomposition of learning and search techniques. *: reference for drop estimation.

| Training Stage | | Solving Stage | | | TSP-50 | | TSP-100 | | TSP-500 | |
|---|---|---|---|---|---|---|---|---|---|---|
| Learning Functionality | Training Objective | Learning Search | Construction | Impr. Search | Drop (%)↓ | Time | Drop (%)↓ | Time | Drop (%)↓ | Time |
| – | – | – | PyConcorde (Applegate et al., 2006) | | 0.000±0.000* | 73.6ms | 0.000±0.000* | 0.4s | 0.000±0.000* | 18.7s |
| – | – | – | LKH3 (trials=500) (Helsgaun, 2017) | | 0.000±0.000 | 0.2s | 0.002±0.019 | 0.2s | 0.322±0.174 | 0.7s |
| – | – | – | LKH3 (trials=5k) (Helsgaun, 2017) | | – | – | 0.000±0.006 | 1.6s | 0.072±0.078 | 4.2s |
| – | – | – | LKH3 (trials=50k) (Helsgaun, 2017) | | – | – | – | – | 0.014±0.029 | 26.3s |
| Edge Pred | SL + Node Norm | – | Greedy | MCTS | 0.021±0.114 | 7.4ms | 0.080±0.261 | 16.3ms | 0.578±0.420 | 1.3s |
| Edge Pred | SL | – | Random (n=50) | MCTS | 0.002±0.025 | 62.5ms | **0.001±0.012** | 0.4s | 0.301±0.153 | 55.2s |
| Edge Pred. | SL + Node Norm | – | Random (n=50) | MCTS | 0.004±0.026 | 50.4ms | 0.002±0.018 | 0.4s | **0.142±0.106** | 55.3s |
| Regret Pred | SL Regret | – | Greedy | Guided LS (t=10s) | 0.004±0.035 | 10.1s | 0.483±0.901 | 10.4s | – | – |
| Regret Pred | SL Regret | – | MCTS Solver (10t) | | 0.001±0.022 | 63.4ms | 1.116±0.781 | 0.2s | – | – |
| Regret Pred | SL Regret | – | Random (n=50) | MCTS | **0.000±0.000** | 0.2s | 0.179±0.241 | 1.8s | – | – |
| Generation | Generative + Sym. | – | Greedy | Two-Opt | 0.046±0.140 | 0.2s | 0.077±0.160 | 0.4s | 0.883±0.543 | 1.6s |
| Generation | Generative + Sym. | Gradient Search | Greedy | Two-Opt | 0.012±0.053 | 0.7s | 0.023±0.068 | 0.8s | 0.445±0.253 | 3.8s |
| Generation | Generative + Sym. | Gradient Search | 8× Greedy | Two-Opt | 0.002±0.024 | 1.3s | 0.003±0.022 | 5.0s | 0.187±0.138 | 28.6s |
| Sequential | RL | – | Sampling + Multistart | – | 0.156±0.265 | 1.3ms | 1.618±0.893 | 1.6ms | – | – |
| Sequential | RL + Sym. Baseline | – | Greedy + Multistart | Two-Opt | 0.122±0.244 | 0.1s | 0.796±0.548 | 0.5s | – | – |
| Sequential | RL + Sym. Baseline | – | Sampling + Multistart | Two-Opt | 0.110±0.242 | 0.2s | 0.729±0.533 | 1.5s | – | – |

Table 8: Generalization Results with models trained on TSP-500. *: reference for drop estimation.

| Training Stage | | Solving Stage | | | TSP-1000 | | TSPLIB 200-1000 | |
|---|---|---|---|---|---|---|---|---|
| Learning Functionality | Training Objective | Learning Search | Construction | Impr. Search | Drop↓ | Time | Drop↓ | Time |
| – | – | – | PyConcorde (Applegate et al., 2006) | | 0.000% | 84.4s | 0.000% | 10.6s |
| Edge Pred. | SL | – | Greedy | MCTS | 1.171% | 6.0s | 1.416% | 1.8s |
| Edge Pred. | SL + Node Norm | – | Greedy | MCTS | 0.798% | 6.1s | 1.252% | 1.8s |
| Edge Pred. | SL + Node Norm | – | Sampling (n=10) | MCTS | 0.687% | 83.5s | **0.517%** | 17.9s |
| Edge Pred. | SL + Node Norm | – | Random (n=10) | MCTS | **0.603%** | 54.6s | 0.678% | 17.3s |
| Solution Generation | Generative | – | Greedy | Two-Opt | 1.811% | 2.3s | 2.431% | 1.6s |
| Solution Generation | Generative + Sym. | – | Greedy | Two-Opt | 1.735% | 1.9s | 1.880% | 1.7s |
| Solution Generation | Generative + Sym. | Gradient Search | Greedy | Two-Opt | 1.293% | 6.1s | 1.337% | 4.4s |
| Solution Generation | Generative + Sym. | – | 8×Greedy | Two-Opt | 1.153% | 22.4s | 1.013% | 5.8s |

models, excel in constructive solvers without heavy reliance on heavy search, achieving 0.002%, 0.003%, 0.188% for TSP-50, 100 and 500. This highlights the potential for future development of more powerful solvers grounded in generative objectives and diffusion backbones. Considering the trade-off between solution quality and solving time, the best solutions are often achieved at the cost of higher computational expense. For comprehensiveness, Table 7 has also selected, for each learning functionality, recompositions at different speed levels, including the fastest approach with stable solution quality for demonstration. This is to accommodate scenarios that prioritize speed or approximation quality over strict optimality, where RL methods stand out with the fastest speed.

**Evaluation for Generalization.** We reconduct ablations over top-performing models to verify the generalization performance of the framework and whether the claims still hold for generalization to larger scale TSP, i.e., TSP-1000, and TSPLIB instances with 200-1000 nodes. The verified models are trained on TSP-500. Comparing methods in Table 8, we observe that the advantages of design principles such as joint probability estimation, symmetry solution representation, online optimization, and search friendliness persist across generalization. Additionally, the newly recomposed search method, *Random_MCTS*, continues to demonstrate a performance advantage in generalization.

Regarding problem size, as observed in Table 9, the general trend is that the larger the scale, the worse the model's relative solving performance. We also find that learning strategies that play supportive roles (e.g. guiding MCTS) are less scale-sensitive, while end-to-end methods (e.g. generative and RL methods) are more affected. This may be because strong search methods mitigate the risk of reduced prediction quality by learning. However, we can also see that methods like generative models still perform well as the scale increases. On the other hand, when considering model complexity and training time, methods with shorter training times, like unsupervised learning, are relatively less affected by scale. This may be because the bottleneck is the model capacity or the amount of learnable information using unsupervised learning rather than task difficulty.

## 5 CONCLUSION AND FUTURE WORK

We design a principled modular framework that integrates existing ML4TSP practices, allowing for inspecting the current system. Based on this unification, we analyze the relationship between learning and solving objectives and show that learning generally improves solving with compatible search algorithms. We also analyze the effectiveness of current ML4TSP efforts, identifying desirable design principles for ML solvers like joint probability estimation, symmetry solution representation, and online optimization. Additionally, we offer insights into how model predictions intervene with strong search methods like MCTS and highlight preferences in learning paradigms for ML4TSP. Leveraging our findings, we propose enhancements to existing methods to compensate for their missing attributes and recompose within the established technique library to establish a factory of learning-based solvers, among which the optimal compositions achieve clear performance gains. Our implementation is carefully organized, facilitating easy development for further research and engineering practices.

## ETHICS STATEMENT

This research complies with ethical standards, utilizing datasets that are either synthetic or publicly available and contain no sensitive or personally identifiable information. It involves no direct human subjects, nor does it present privacy or security concerns. All methodologies and experiments were conducted in line with legal regulations and established research integrity practices. There are no conflicts of interest, external sponsorship influences, or concerns regarding discrimination, bias, or fairness. Moreover, the research does not lead to any harmful insights or applications.

## REPRODUCIBILITY STATEMENT

We have taken steps to ensure the reproducibility of the results presented in this paper. The experimental settings, including datasets and model designs, are thoroughly described in Section 4. Additional details, such as algorithm details and model configurations, are provided in Appendix F. We provide full records of our experiments in Table 9 and provide minimalistic code example for the implementation in Appendix D. Source code will be made publicly available upon acceptance.

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

# Appendix

CONTENTS

Table 9: Full Results of Learning and Searching Technique Recomposition for ML4TSP Solvers.

| Method | Training Stage | Solving Stage | | | TSP-50 | | | TSP-100 | | | TSP-500 | | |
|---|---|---|---|---|---|---|---|---|---|---|---|---|---|
| | Training Objective | Learning Search | Construction | Impr. Search | Length↓ | Drop↓ | Time | Length↓ | Drop↓ | Time | Length↓ | Drop↓ | Time |
| *Traditional Solver* | | | | | | | | | | | | | |
| Concorde (Applegate et al., 2006) | – | – | – | – | 5.688* | 0.000% | 73.6ms | 7.756* | 0.000% | 0.404s | 16.546* | 0.000% | 18.672s |
| Greedy | – | – | – | – | 7.014 | 23.301% | 0.31ms | 9.680 | 24.798% | 1.15ms | 20.808 | 25.766% | 35.84ms |
| Insertion | – | – | – | – | 6.126 | 7.708% | 0.15ms | 8.508 | 9.699% | 0.28ms | 18.608 | 12.461% | 5.75ms |
| GA-EAX Nagata & Kobayashi (2013) | – | – | – | – | 5.688 | 0.000% | 0.10s | 7.756 | 0.000% | 0.18s | 16.546 | 0.001% | 1.86s |
| LKH3 (trials=500) Helsgaun (2017) | – | – | – | – | 5.688 | 0.000% | 0.177s | 7.756 | 0.002% | 0.216s | 16.599 | 0.322% | 0.703s |
| LKH3 (trials=5k) Helsgaun (2017) | – | – | – | – | | | | 7.756 | 0.000% | 1.561s | 16.558 | 0.072% | 4.172s |
| LKH3 (trials=50k) Helsgaun (2017) | – | – | – | – | – | – | – | – | – | – | 16.548 | 0.014% | 26.344s |
| *Edge Prediction* | | | | | | | | | | | | | |
| GCN (Joshi et al., 2019) | SL | – | Greedy | – | 5.776 | 1.530% | 7.6ms | 8.307 | 7.078% | 10.6ms | 18.571 | 12.237% | 0.111s |
| ML4TSP | SL | – | Greedy | Two-Opt | 5.694 | 0.115% | 7.6ms | 7.806 | 0.649% | 12.0ms | 16.891 | 2.089% | 0.128s |
| ML4TSP | SL | – | Greedy | Guided LS (t=1) | 5.723 | 0.614% | 1.084s | 8.014 | 3.319% | 1.236s | 17.818 | 7.682% | 45.469s |
| ML4TSP | SL | – | Greedy | MCTS | 5.689 | 0.018% | 8.4ms | 7.768 | 0.154% | 16.8ms | 16.684 | 0.838% | 1.320s |
| GCN (Joshi et al., 2019) | SL | – | Beam (n=1280) | – | 5.689 | 0.023% | 1.099s | 7.808 | 0.673% | 2.320s | 20.456 | 23.629% | 11.602s |
| ML4TSP | SL | – | Beam (n=1280) | Two-Opt | 5.688 | 0.015% | 1.099s | 7.767 | 0.149% | 2.332s | 16.948 | 2.431% | 12.859s |
| ML4TSP | SL | – | Beam (n=10) | MCTS | 5.688 | 0.008% | 38.8ms | 7.758 | 0.027% | 0.102s | 16.609 | 0.380% | 11.409s |
| ML4TSP | SL | – | Beam (n=20) | MCTS | 5.688 | 0.006% | 54.9ms | 7.757 | 0.014% | 0.176s | 16.598 | 0.315% | 22.712s |
| ML4TSP | SL | – | Beam (n=50) | MCTS | 5.688 | 0.005% | 0.111s | 7.757 | 0.009% | 0.475s | 16.585 | 0.235% | 56.286s |
| ML4TSP | SL | – | Sampling (n=10) | MCTS | 5.688 | 0.005% | 37.5ms | 7.756 | 0.008% | 0.124s | 16.645 | 0.597% | 11.789s |
| ML4TSP | SL | – | Sampling (n=20) | MCTS | 5.688 | 0.004% | 61.5ms | 7.756 | 0.004% | 0.280s | 16.630 | 0.509% | 23.338s |
| ML4TSP | SL | – | Sampling (n=50) | MCTS | 5.688 | 0.002% | 0.114s | 7.756 | 0.000% | 0.549s | 16.611 | 0.396% | 58.122s |
| ML4TSP | SL | – | Random (n=10) | MCTS | 5.688 | 0.004% | 17.5ms | 7.757 | 0.011% | 81.4ms | 16.632 | 0.519% | 11.159s |
| ML4TSP | SL | – | Random (n=20) | MCTS | 5.688 | 0.003% | 28.7ms | 7.756 | 0.003% | 0.155s | 16.616 | 0.422% | 22.131s |
| ML4TSP | SL | – | Random (n=50) | MCTS | 5.688 | 0.002% | 62.5ms | 7.756 | 0.001% | 0.374s | 16.596 | 0.301% | 55.195s |
| Att-GCN (Fu et al., 2021) | SL | – | | MCTS Solver (1t) | 5.688 | 0.009% | 11.7ms | 7.759 | 0.035% | 29.9ms | 16.704 | 0.958% | 1.078s |
| Att-GCN (Fu et al., 2021) | SL | – | | MCTS Solver (5t) | 5.688 | 0.007% | 31.7ms | 7.757 | 0.009% | 0.118s | 16.634 | 0.530% | 5.082s |
| Att-GCN (Fu et al., 2021) | SL | – | | MCTS Solver (10t) | 5.688 | 0.006% | 56.7ms | 7.756 | 0.006% | 0.217s | 16.617 | 0.433% | 10.081s |
| ML4TSP | SL + Node Norm | – | Greedy | – | 5.727 | 0.674% | 7.4ms | 7.968 | 2.721% | 10.1ms | 17.952 | 8.496% | 0.111s |
| ML4TSP | SL + Node Norm | – | Greedy | Two-Opt | 5.692 | 0.082% | 7.4ms | 7.779 | 0.292% | 11.7ms | 16.765 | 1.325% | 0.128s |
| ML4TSP | SL + Node Norm | – | Greedy | MCTS | 5.689 | 0.021% | 7.4ms | 7.762 | 0.080% | 16.3ms | 16.641 | 0.578% | 1.297s |
| ML4TSP | SL + Node Norm | – | Beam (n=1280) | – | 5.688 | 0.016% | 1.132s | 7.798 | 0.545% | 2.241s | 20.133 | 21.678% | 13.720s |
| ML4TSP | SL + Node Norm | – | Beam (n=1280) | Two-Opt | 5.688 | 0.012% | 1.142s | 7.768 | 0.156% | 2.254s | 16.908 | 2.189% | 14.009s |
| ML4TSP | SL + Node Norm | – | Beam (n=10) | MCTS | 5.688 | 0.012% | 28.1ms | 7.758 | 0.027% | 98.6ms | 16.585 | 0.238% | 11.417s |
| ML4TSP | SL + Node Norm | – | Beam (n=20) | MCTS | 5.688 | 0.011% | 56.7ms | 7.757 | 0.019% | 0.179s | 16.580 | 0.208% | 22.652s |
| ML4TSP | SL + Node Norm | – | Beam (n=50) | MCTS | 5.688 | 0.008% | 94.5ms | 7.757 | 0.013% | 0.421s | 16.571 | 0.152% | 56.094s |
| ML4TSP | SL + Node Norm | – | Sampling (n=10) | MCTS | 5.688 | 0.009% | 28.1ms | 7.757 | 0.011% | 0.108s | 16.590 | 0.265% | 11.882s |
| ML4TSP | SL + Node Norm | – | Sampling (n=20) | MCTS | 5.688 | 0.008% | 49.6ms | 7.756 | 0.005% | 0.207s | 16.581 | 0.214% | 23.580s |
| ML4TSP | SL + Node Norm | – | Sampling (n=50) | MCTS | 5.688 | 0.006% | 0.115s | 7.756 | 0.003% | 0.529s | 16.569 | 0.142% | 59.119s |
| ML4TSP | SL + Node Norm | – | Random (n=10) | MCTS | 5.688 | 0.007% | 16.5ms | 7.756 | 0.009% | 81.5ms | 16.586 | 0.245% | 11.184s |
| ML4TSP | SL + Node Norm | – | Random (n=20) | MCTS | 5.688 | 0.004% | 27.7ms | 7.756 | 0.004% | 0.156s | 16.579 | 0.200% | 22.308s |
| ML4TSP | SL + Node Norm | – | Random (n=50) | MCTS | 5.688 | 0.004% | 50.4ms | 7.756 | 0.002% | 0.389s | 16.569 | 0.142% | 55.294s |
| ML4TSP | SL + Node Norm | – | | MCTS Solver (10t) | 5.688 | 0.010% | 56.7ms | 7.760 | 0.013% | 0.210s | 16.600 | 0.329% | 10.188s |
| ML4TSP | UL | – | Greedy | – | 6.405 | 12.600% | 11.4ms | 8.870 | 14.360% | 17.0ms | – | – | – |
| ML4TSP | UL | – | Greedy | Two-Opt | 5.846 | 2.785% | 12.3ms | 8.003 | 3.182% | 35.0ms | – | – | – |
| ML4TSP | UL | – | Greedy | MCTS | 5.863 | 3.096% | 28.9ms | 8.069 | 4.039% | 89.0ms | – | – | – |
| UTSP (Min et al., 2023) | UL | – | | MCTS Solver (10t) | 5.818 | 2.300% | 62.9ms | 8.069 | 4.463% | 0.223s | – | – | – |
| ML4TSP | RL | – | Sampling | – | 5.985 | 5.238% | 11.4ms | 8.481 | 9.364% | 94.5ms | 19.296 | 16.612% | 1.039s |
| ML4TSP | RL | Active Search | Sampling | – | 5.863 | 3.086% | 3.234s | 8.084 | 4.234% | 6.703s | 17.865 | 7.959% | 258.281s |
| DIMES (Qiu et al., 2022) | Meta + RL | – | Greedy | – | 6.360 | 11.847% | 15.6ms | 8.790 | 13.339% | 18.0ms | 19.605 | 18.489% | 0.375s |
| DIMES (Qiu et al., 2022) | Meta + RL | – | Greedy | Two-Opt | 5.933 | 4.310% | 35.9ms | 8.135 | 4.885% | 0.124s | 18.706 | 13.056% | 1.305s |
| DIMES (Qiu et al., 2022) | Meta + RL | – | | MCTS Solver (10t) | 5.697 | 0.172% | 60.6ms | 7.807 | 0.661% | 0.216s | 17.425 | 5.318% | 10.296s |
| DIMES (Qiu et al., 2022) | Meta + RL | Active Search | Greedy | – | 5.897 | 3.681% | 4.230s | 8.111 | 4.583% | 8.575s | 17.763 | 7.341% | 264.781s |
| DIMES (Qiu et al., 2022) | Meta + RL | Active Search | Sampling | – | 5.859 | 3.006% | 2.884s | 8.061 | 3.940% | 8.508s | 17.694 | 6.924% | 274.797s |
| ML4TSP | Meta + RL | – | Greedy | Two-Opt | 5.823 | 2.387% | 18.0ms | 8.007 | 3.232% | 57.0ms | 17.165 | 3.742% | 0.453s |
| ML4TSP | Meta + RL | – | Greedy | LS | 6.016 | 5.756% | 65.6ms | 8.339 | 7.520% | 0.457s | 17.938 | 8.414% | 48.688s |
| ML4TSP | Meta + RL | – | Greedy | Guided LS (t=1s) | 5.986 | 5.231% | 1.112s | 8.335 | 7.463% | 1.382s | 17.938 | 8.408% | 51.406s |
| ML4TSP | Meta + RL | – | Greedy | MCTS | 5.765 | 1.337% | 18.8ms | 7.909 | 1.973% | 42.4ms | 17.354 | 4.884% | 1.385s |
| ML4TSP | Meta + RL | – | Random (n=50) | MCTS | 5.688 | 0.018% | 0.241s | 7.765 | 0.123% | 1.522s | 17.655 | 6.703% | 59.297s |
| ML4TSP | Meta + RL | – | Sampling | Two-Opt | 5.791 | 1.821% | 0.141s | 7.979 | 2.880% | 0.232s | 17.578 | 6.228% | 6.406s |
| ML4TSP | Meta + RL | Active Search | Sampling | Two-Opt | 5.776 | 1.549% | 4.414s | 7.925 | 2.176% | 8.859s | 17.263 | 4.322% | 277.031s |
| ML4TSP | Meta + RL | Active Search | Sampling+RandStart | – | 5.735 | 0.823% | 4.862s | 7.918 | 2.089% | 9.330s | 17.646 | 6.635% | 272.813s |
| *Edge Regret Prediction* | | | | | | | | | | | | | |
| ML4TSP | SL Regret | – | Greedy | – | 6.179 | 8.651% | 18.8ms | 8.966 | 15.597% | 16.4ms | – | – | – |
| ML4TSP | SL Regret | – | Greedy | – | 5.707 | 0.333% | 19.5ms | 7.857 | 1.295% | 0.129s | – | – | – |
| ML4TSP | SL Regret | – | Greedy | LS | 5.784 | 1.695% | 0.064s | 8.127 | 4.786% | 0.306s | – | – | – |
| GNNGLS (Hudson et al., 2022) | SL Regret | – | Greedy | Guided LS (t=1s) | 5.693 | 0.098% | 1.080s | 7.837 | 1.049% | 1.389s | – | – | – |
| GNNGLS (Hudson et al., 2022) | SL Regret | – | Greedy | Guided LS (t=10s) | 5.688 | 0.004% | 10.078s | 7.793 | 0.483% | 10.359s | – | – | – |
| ML4TSP | SL Regret | – | Greedy | MCTS | 5.702 | 0.255% | 17.2ms | 7.927 | 2.210% | 54.0ms | – | – | – |
| ML4TSP | SL Regret | – | Random (n=50) | MCTS | 5.688 | 0.000% | 0.207s | 7.770 | 0.179% | 1.843s | – | – | – |
| ML4TSP | SL Regret | – | | MCTS Solver (10t) | 5.688 | 0.001% | 63.4ms | 7.843 | 1.116% | 0.218s | – | – | – |
| *Solution Generation* | | | | | | | | | | | | | |
| DIFUSCO (Sun & Yang, 2023) | Generative | – | Greedy | – | 5.709 | 0.384% | 0.388s | 7.845 | 1.136% | 0.409s | 18.144 | 9.639% | 1.188s |
| DIFUSCO (Sun & Yang, 2023) | Generative | – | Greedy | Two-Opt | 5.694 | 0.105% | 0.394s | 7.776 | 0.261% | 0.409s | 16.800 | 1.524% | 1.203s |
| DIFUSCO (Sun & Yang, 2023) | Generative | – | 8× Greedy | – | 5.688 | 0.014% | 0.688s | 7.761 | 0.062% | 2.355s | 17.312 | 4.613% | 6.867s |
| DIFUSCO (Sun & Yang, 2023) | Generative | – | 8× Greedy | Two-Opt | 5.688 | 0.010% | 0.689s | 7.759 | 0.035% | 2.356s | 16.759 | 1.288% | 6.914s |
| T2T (Li et al., 2023b) | Generative | Gradient Search | Greedy | – | 5.690 | 0.039% | 1.164s | 7.788 | 0.127% | 1.198s | 17.444 | 5.414% | 2.805s |
| T2T (Li et al., 2023b) | Generative | Gradient Search | Greedy | Two-Opt | 5.689 | 0.022% | 1.181s | 7.761 | 0.065% | 1.217s | 16.695 | 0.885% | 2.836s |
| ML4TSP | Generative + Sym. | – | Greedy | – | 5.692 | 0.076% | 0.229s | 7.770 | 0.186% | 0.398s | 17.654 | 6.695% | 1.578s |
| ML4TSP | Generative + Sym. | – | Greedy | Two-Opt | 5.690 | 0.046% | 0.230s | 7.762 | 0.077% | 0.400s | 16.692 | 0.883% | 1.586s |
| ML4TSP | Generative + Sym. | – | Greedy | MCTS | 5.690 | 0.040% | 0.318s | 7.762 | 0.080% | 0.489s | 16.689 | 0.863% | 1.716s |
| ML4TSP | Generative + Sym. | – | 8× Greedy | – | 5.688 | 0.007% | 0.701s | 7.757 | 0.011% | 2.450s | 16.785 | 1.443% | 10.328s |
| ML4TSP | Generative + Sym. | – | 8× Greedy | Two-Opt | 5.688 | 0.006% | 0.702s | 7.757 | 0.009% | 2.450s | 16.599 | 0.320% | 10.375s |
| ML4TSP | Generative + Sym. | – | 8× Greedy | MCTS | 5.688 | 0.006% | 0.859s | 7.757 | 0.009% | 3.141s | 16.605 | 0.357% | 21.734s |
| ML4TSP | Generative + Sym. | – | | MCTS Solver (10t) | 5.691 | 0.051% | 0.446s | 7.763 | 0.096% | 1.002s | 17.027 | 2.910% | 11.510s |
| ML4TSP | Generative + Sym. | – | Random (n=50) | MCTS | 5.689 | 0.032% | 0.439s | 7.761 | 0.068% | 3.916s | 16.619 | 0.443% | 56.554s |
| ML4TSP | Generative + Sym. | Gradient Search | Greedy | – | 5.688 | 0.015% | 0.717s | 7.758 | 0.023% | 0.825s | 16.989 | 2.675% | 3.711s |
| ML4TSP | Generative + Sym. | Gradient Search | Greedy | Two-Opt | 5.688 | 0.012% | 0.718s | 7.758 | 0.023% | 0.827s | 16.619 | 0.455% | 3.750s |
| ML4TSP | Generative + Sym. | Gradient Search | Greedy | MCTS | 5.688 | 0.018% | 1.208s | 7.759 | 0.041% | 0.827s | 16.621 | 0.445% | 3.914s |
| ML4TSP | Generative + Sym. | Gradient Search | 8× Greedy | – | 5.688 | 0.002% | 1.317s | 7.756 | 0.003% | 4.406s | 16.613 | 0.404% | 26.898s |
| ML4TSP | Generative + Sym. | Gradient Search | 8× Greedy | Two-Opt | 5.688 | 0.002% | 1.327s | 7.756 | 0.003% | 4.453s | 16.577 | 0.187% | 27.094s |
| ML4TSP | Generative + Sym. | Gradient Search | 8× Greedy | MCTS | 5.688 | 0.002% | 1.380s | 7.756 | 0.003% | 4.969s | 16.599 | 0.323% | 28.594s |
| *Sequential Node Prediction* | | | | | | | | | | | | | |
| AM (Kool et al., 2018) | RL | – | Greedy | – | 5.747 | 1.040% | 0.1ms | 7.951 | 2.520% | 0.2ms | – | – | – |
| ML4TSP | RL | – | Greedy | Two-Opt | 5.733 | 0.799% | 1.0ms | 7.902 | 1.887% | 3.6ms | – | – | – |
| ML4TSP | RL | – | Greedy + Multistart | – | 5.732 | 0.782% | 0.3ms | 7.920 | 2.110% | 1.2ms | – | – | – |
| ML4TSP | RL | – | Greedy + Multistart | Two-Opt | 5.697 | 0.167% | 48.4ms | 7.826 | 0.898% | 0.438s | – | – | – |
| AM (Kool et al., 2018) | RL | – | Sampling | – | 5.747 | 1.212% | 0.1ms | 7.977 | 2.846% | 0.2ms | – | – | – |
| ML4TSP | RL | – | Sampling | Two-Opt | 5.735 | 0.838% | 1.117ms | 7.906 | 1.927% | 4.3ms | – | – | – |
| ML4TSP | RL | – | Sampling + Multistart | – | 5.730 | 0.739% | 0.4ms | 7.918 | 2.092% | 1.2ms | – | – | – |
| ML4TSP | RL | – | Sampling + Multistart | Two-Opt | 5.694 | 0.113% | 55.5ms | 7.827 | 0.919% | 1.248s | – | – | – |
| POMO (Kwon et al., 2020) | RL + Multistart | – | Greedy | – | 5.698 | 0.179% | 1.7ms | 7.883 | 1.641% | 1.6ms | – | – | – |
| ML4TSP | RL + Multistart | – | Greedy | Two-Opt | 5.693 | 0.103% | 16.4ms | 7.854 | 1.261% | 76.6ms | – | – | – |
| POMO (Kwon et al., 2020) | RL + Multistart | – | Sampling | – | 5.697 | 0.156% | 1.3ms | 7.881 | 1.618% | 1.6ms | – | – | – |
| ML4TSP | RL + Multistart | – | Sampling | Two-Opt | 5.693 | 0.087% | 18.0ms | 7.847 | 1.172% | 0.161s | – | – | – |
| Sym-NCO (Kim et al., 2022) | RL + Sym. Baseline | – | Greedy | – | 5.738 | 0.877% | 0.6ms | 7.927 | 2.209% | 0.6ms | – | – | – |
| ML4TSP | RL + Sym. Baseline | – | Greedy | Two-Opt | 5.726 | 0.680% | 1.5ms | 7.891 | 1.745% | 3.8ms | – | – | – |
| ML4TSP | RL + Sym. Baseline | – | Greedy + Multistart | – | 5.719 | 0.549% | 0.9ms | 7.901 | 1.869% | 1.7ms | – | – | – |
| ML4TSP | RL + Sym. Baseline | – | Greedy + Multistart | Two-Opt | 5.695 | 0.122% | 0.120s | 7.818 | 0.796% | 0.543s | – | – | – |
| Sym-NCO (Kim et al., 2022) | RL + Sym. Baseline | – | Sampling | – | 5.747 | 1.041% | 0.6ms | 7.957 | 2.593% | 0.6ms | – | – | – |
| ML4TSP | RL + Sym. Baseline | – | Sampling | Two-Opt | 5.729 | 0.726% | 1.7ms | 7.901 | 1.872% | 4.4ms | – | – | – |
| ML4TSP | RL + Sym. Baseline | – | Sampling + Multistart | – | 5.717 | 0.512% | 0.9ms | 7.898 | 1.838% | 1.7ms | – | – | – |
| ML4TSP | RL + Sym. Baseline | – | Sampling + Multistart | Two-Opt | 5.694 | 0.110% | 0.173s | 7.813 | 0.729% | 1.502s | – | – | – |

# A  GENERAL DISCUSSION OF DESIGN CHOICES

With the recomposition results, we obtain the following observations regarding the general design choices for ML4TSP solvers: 1) MCTS variants can significantly enhance the performance, while it is not universally compatible with all learning designs. For adaptability, smoother and more continuous predictions tend to combine more easily with MCTS. With MCTS, the optimal combination in our framework is achieved when learning the solution proximity objective with node-wise normalization for edge prediction. This combination, together with the random MCTS solving strategy, achieves

drops of 0.000%, 0.001%, and 0.142% on TSP of sizes 50, 100, and 500, respectively. 2) Without relying on tailored heuristics like MCTS, the generative distribution proximity objective based on the symmetry representation achieves the optimal results when adopting gradient search, $8\times$ Greedy, and Two-Opt for the solving stage, achieving drops of 0.002%, 0.003%, and 0.187% on TSP of sizes 50, 100, and 500, respectively. 3) Regarding the paradigms of learning-based solvers, we discover a clear performance advantage of non-autoregressive methods to autoregressive sequence models, and supervised methods are significantly ahead of unsupervised methods, especially without the support of powerful heuristic search. For scalability, we notice that acquiring supervision for edge regret is extremely time-consuming, making it impractical for solving larger-scale problems like TSP-500. Meanwhile, RL-based sequence models also face challenges on larger-scale problems due to issues like sparse rewards and training instability, which also struggle to support training on larger-scale instances.

## B    MORE DISCUSSION OF RELATED WORK

Learning-based designs have advanced CO (Bengio et al., 2021; Cappart et al., 2021; Peng et al., 2021) leveraging graph learning (Bao et al., 2024; 2025; Wu et al., 2023). TSP has long been studied by the machine learning community since early works like (Vinyals et al., 2015; Khalil et al., 2017) and is one of the most extensively studied CO problems. Other CO problems like MIS (Brusca et al., 2023; Goshvadi et al., 2024; Wang & Li, 2023), CVRP (Ma et al., 2024; Hou et al., 2023; Kim et al., 2023) are also widely studied, which can sometimes share similar methodology designs. CO problems have found widespread applications in areas such as chip design (Cheng et al., 2022a), molecule generation (Yang et al., 2024; Wu et al., 2024), and cryptography (Zheng et al., 2024).

Learning-based CO solvers involve autoregressive methods (Khalil et al., 2017; Kool et al., 2018; Kwon et al., 2020; Hottung et al., 2021a; Kim et al., 2022) that directly construct solutions by sequentially determining decision variables until a complete solution is constructed, and non-autoregressive methods (Joshi et al., 2019; Fu et al., 2021; Geisler et al., 2022; Qiu et al., 2022; Sun & Yang, 2023; Zheng et al., 2024) that predict soft-constrained solutions in one shot and then perform post-processing to achieve feasibility. The neural predictions can also be leveraged to guide the local search process, which iteratively refines a given solution (d O Costa et al., 2020; Wu et al., 2021; Chen & Tian, 2019; Li et al., 2021; Fu et al., 2021; Hudson et al., 2022). Recently, quantum machine learning has shown potential in solving these NP problems empowered or inspired by quantum computing properties (Ye et al., 2023; 2025).

Generative models (Li et al., 2022; 2023c) have been leveraged in CO to resolve data bottlenecks (Li et al., 2023a; Che et al., 2024; Guo et al., 2024) and have also shown promise in solving CO thanks to their strong representational capabilities and informative distribution estimation. It models the problem-solving task as a conditional generation task for learning solution distributions conditioned on given instances (Hottung et al., 2021a; Cheng et al., 2022b; Sun & Yang, 2023; Li et al., 2023b; Sanokowski et al., 2024; Li et al., 2024). The development line of diffusion-based solvers has achieved the current SOTA performance in neural CO solving. DIFUSCO (Sun & Yang, 2023) adapts the discrete diffusion models (Austin et al., 2021; Hoogeboom et al., 2021) in CO and has attained competitive performance in solving TSP and MIS. The T2T framework (Li et al., 2023b) further introduces an objective-guided gradient search process in the diffusion model during solving to leverage the learned distribution and has achieved the SOTA performance. DiffUCO (Sanokowski et al., 2024) optimizes the diffusion-based solver through an unsupervised learning approach. Recently, Fast T2T (Li et al., 2024) proposes a more powerful optimization consistency model for solving CO problems that achieves tens of times speedup compared to previous raw diffusion-based solvers.

## C    ADAPTATION OF THE FRAMEWORK TO OTHER CO PROBLEMS

### C.1    GUIDE FOR CROSS-PROBLEM ADAPTATION

The ML field has gone the farthest optimizing the TSP, and we consider TSP as one of the most extensively studied CO problems with the richest learning and searching techniques. Note numerous methods have been devised in ML4CO. While some of the best-performing methods show promising results, they still exhibit a gap compared to traditional methods. The very first motivation of this work is to reassess the role of learning for CO (or specifically TSP) and to discern which parts of existing

techniques are genuinely beneficial and which are not. For this purpose, we select TSP to incorporate and analyze learning and searching techniques as comprehensively as possible and, to the maximum extent, retain the potential for application to other problems.

This framework can be customized for other CO problems with the same modular organization and learning-based designs. The primary adaptions required would involve adjusting the optimization variables, e.g. replacing edges with nodes for node-decision problems such as Maximum Independent Set (MIS), as well as modifying the (optional) heuristic search designs that are problem-specific. Specifically, the pipeline for a new problem can be still generally be organized by a prediction-and-seach pipeline like TSP, where the learning models can be trained with learning tasks including 1) decision variable prediction, 2) variable regret prediction, 3) solution generation, and 4) sequential variable prediction, as well as training objectives including 1) supervised solution proximity, 2) unsupervised solving objective, 3) reinforcement solving objective, 4) supervised variable regret proximity, 5) solution distribution proximity, potentially with the support of 6) meta learning objective. Then in the solving stage, the neural prediction can be further improved by learning-based search methods for specific problems, and the search phase can follow the construction-search pipeline with constructive search like Greedy and improvement-based search like local search.

Additionally, many parts of our analysis, e.g., the study of the relationship between learning and solving objectives, the benefits of satisfying certain design principles like joint probability estimation, symmetry solution representation, online optimization, etc., indeed study the general learning designs and its synergy with search methods, maintaining the potential to have applicability beyond TSP in design. For example, for MIS, we can also consider achieving joint probability estimation by introducing a normalization term on the predicted node probabilities to enhance performance. For other problems with sequential solutions, we can consider enforcing the symmetry property for solution representations. In general pipeline designs, we can also consider introducing optimization during the testing phase to enhance performance.

## C.2 ML4CO-Kit Library: Meta Framework

To facilitate adaptation to other CO problems, we have developed a task-agnostic skeleton and common toolkit, which provides robust support for the development of learning-based solvers across different tasks. This toolkit focuses on facilitating method development while leaving room for customizations of problem-specific implementations. Key features include:

1. A generic skeleton for organizing ML4CO frameworks;
2. Base classes to streamline method development;
3. Traditional solver baselines and reference solution acquisition;
4. Data generation for various distributions;
5. Problem and solution visualization tools;
6. Evaluators for multiple problem types;
7. Common post-processing algorithms.

This toolkit enables the straightforward adaptation of our framework to new problems by inheriting base classes and adhering to the pre-defined project structure. The source code is publicly available at https://github.com/Thinklab-SJTU/ML4CO-Kit. Based on this toolkit, we extend our framework design to CVRP and MIS problems, which include foundational classes for developing learning-based CVRP and MIS solvers. These frameworks currently implement several popular learning-based solvers and can be expanded to integrate additional methods using the same modular structure.

## C.3 Empirical Results

This subsection verifies the effectiveness of satisfying the discovered design principles in TSP to show the reference value of the ML4TSP analysis to other CO problems. Here, we verify the Capacitated Vehicle Routing Problem (VRP) and Maximum Independent Set (MIS) problems. Building upon the principles, we introduce novel designs seamlessly integrated into existing pipelines to boost solving performance and validate the effectiveness of adhering to these principles throughout the process.

### C.3.1 Experimental Setup

**CVRP Datasets.** Following Ma et al. (2024), we use samples with coordinates that are randomly uniformly distributed, demands that are random integers between 1 and 9, and a total capacity of 40 as our training and testing sets. The training set size is 128000, with 100 training epochs. The testing set size is 10000, consistent with Ma et al. (2024). Note that when using neural networks for inference, the batch size is set to 100.

**CVRP Metrics.** We adopt three evaluation metrics to measure performance: 1) Length: the average total distance of the solved tours w.r.t. the corresponding instances; 2) Drop: the relative performance drop w.r.t. length compared to the nearly optimal solutions (obtained by the heuristic solver HGS (Vidal, 2022) in this paper); 3) Time: the average computational time per instance.

**MIS Datasets.** We use SATLIB[2] as the dataset. Following Qiu et al. (2022); Sun & Yang (2023); Li et al. (2023b), we use 39500 samples as the training set and 500 samples as the test set.

**MIS Metrics.** Following Qiu et al. (2022); Sun & Yang (2023); Li et al. (2023b), we adopt three evaluation metrics to measure model performance: 1) Size: the average size of the solutions w.r.t. the corresponding instances, i.e. the objective. 2) Drop: the relative performance drop w.r.t. size compared to the nearly optimal solutions (obtained by the heuristic solver KaMIS (Lamm et al., 2016) in this paper); 3) Time: the average computational time per instance.

### C.3.2 Joint Probability Estimation Helps Capture Variable Correlations

This subsection verifies whether the joint probability estimation property also benefits other CO problems. Here, we adopt the MIS problem for evaluation. For MIS, the decision variable is the nodes in the given graph. In the learning functionality of node prediction, the *SL* training objective generally enforces supervision over each node's binary classification results. This is based on the assumption of mutual statistical independence of the individual solution parameters, such that the solution can be represented by a product of Bernoulli distributions for the individual solution variables (Sanokowski et al., 2023). However, since the CO solutions are highly constrained with strong correlations among the solution variables, such an assumption can lead to a performance drop.

With GNN raw predictions, which correspond to $n$ nodes, prior methods utilize the softmax operation over the last dimension to ensure that the probabilities of positive and negative edges sum up to 1. The cross-entropy loss involves $n$ nodes' binary classification, lacking a general capture of joint probability over the solution variables. To this end, we propose a relaxed probability normalization term achieved by LinSAT layer Wang et al. (2023), which projects the raw predicted probabilities to the constrained space, where the positive node probabilities within each node's adjacent node group to sum up to less than the size of the adjacent group minus one. This is a largely relaxed constraint for the MIS problem, yet it provides the chance to connect among the node probabilities.

Under the node prediction functionality, we also include methods of solution distribution proximity objective into comparison, which considers a solution as the whole entity, allowing it to naturally adhere to the joint probability distribution. Table 10 shows the ablation. Methods with joint probability estimation generally achieve lower drops.

Table 10: Joint probability estimation ablation based on methods with Variable Prediction as the learning functionality for MIS.

| Objective | Joint Prob. | Solving Stage | SATLIB | | |
|---|---|---|---|---|---|
| | | | Size↑ | Drop↓ | Time |
| KaMIS (10s) (Lamm et al., 2016) | | | 425.954 | 0.000±0.000% | 10s |
| SL | ✗ | Greedy | 421.692 | 1.004±0.524% | 22.7ms |
| SL + LinSAT Norm | ✔ | Greedy | 422.098 | 0.907±0.473% | 0.545s |
| Generative | ✔ | Greedy | 424.496 | 0.344±0.329% | 1.101s |

---

[2] https://www.cs.ubc.ca/~hoos/SATLIB/Benchmarks/SAT/CBS/descr_CBS.html

### C.3.3 SYMMETRY SOLUTION REPRESENTATION INTRODUCES EQUIVALENCE AWARENESS

Similar to TSP, a CVRP tour holds symmetry or permutation invariance (Kwon et al., 2020). Since CO aims to produce optimal solutions, the neural model is expected to be aware of the equivalences among such sequences and should be optimized to consistently yield the same solution regardless of the variances in forms. This subsection endeavors to introduce the symmetry solution representation attribute to methods lacking symmetry consideration within the framework, thereby enhancing problem-solving performance.

For the *RL* objective, methods decode solutions sequentially during optimization. To achieve symmetry, we refer to Kwon et al. (2020) to optimize over diverse rollouts with multiple starting points as well as use symmetry baselines (Kim et al., 2022) for *Sequential* functionality. These techniques raise the model's awareness of symmetry. Table 11 presents solving results on CVRP-50, showing a notable performance gain by introducing symmetry.

Table 11: Symmetry ablation based on methods with *RL* as the training objective on CVRP.

| Functionality | Symmetry | Solving Stage | CVRP-50 | | |
|---|---|---|---|---|---|
| | | | Length↓ | Drop↓ | Time |
| HGS (1s) (Vidal, 2022) | | | 10.366 | 0.000±0.000% | 1.0s |
| Sequential | ✗ | Greedy | 11.138 | 7.479±2.953% | 0.9ms |
| + MultiStart | ✔ | Greedy | 10.605 | 2.305±1.184% | 2.2ms |
| + Sym. Baseline | ✔ | Greedy | 10.989 | 6.035±2.613% | 0.9ms |
| Sequential | ✗ | Sampling | 11.201 | 8.093±3.187% | 0.9ms |
| + MultiStart | ✔ | Sampling | 10.594 | 2.195±1.102% | 2.2ms |
| + Sym. Baseline | ✔ | Sampling | 11.031 | 6.441±2.705% | 0.9ms |

### C.3.4 ONLINE OPTIMIZATION COMPLEMENTS GENERALIZATION CAPABILITY

In order to bridge the gap between training for average performance and solving for performance in new instances, instance-specific optimization can be helpful. We verify the effectiveness of techniques that optimize in the testing phase in Sec. 3.2.1: the gradient search for methods with *Generation* as the functionality. Table 12 ablates learning-based search, indicating a promising gain with learning-based search, however at the cost of additional computation.

Table 12: Ablation on online optimization property achieved by learning-based search on MIS.

| Functionality | Objective | Solving Stage | SATLIB | | |
|---|---|---|---|---|---|
| | | | Size↑ | Drop↓ | Time |
| Generation | Generative | Greedy | 424.496 | 0.344±0.329% | 1.101s |
| | Generative | Gradient Search + Greedy | 425.006 | 0.224±0.213% | 2.380s |

### C.3.5 SEARCH FRIENDLINESS CONTRIBUTES TO LEARNING-SEARCH SYNERGY

Techniques that consider how to better integrate learning with search can benefit problem-solving. This subsection verifies the effectiveness of search friendliness designs by promoting diversity in the predicted solutions (e.g. modeling solution distribution) and incorporating the search process into optimization (e.g. through meta learning to learn optimal initialization for solving stage). For the effectiveness of prediction diversity, Table 13 demonstrates the improvement in solving performance achieved by adopting meta learning and employing multiple sampling, which is contributed by increased diversity.

Table 13: Ablation on search friendliness property on MIS.

| Functionality | Objective | Search Frnd. | Solving Stage | SATLIB | | |
|---|---|---|---|---|---|---|
| | | | | Size↑ | Drop↓ | Time |
| Variable Prediction | RL | ✗ | Sampling | 408.480 | 4.104±3.558% | 27m34s |
| Variable Prediction | Meta + RL | ✔ | Sampling | 410.596 | 3.607±2.864% | 28m5s |
| Variable Prediction | Generative | ✗ | Greedy | 424.496 | 0.344±0.329% | 1.101s |
| Variable Prediction | Generative | ✔ | 8×Greedy | 425.142 | 0.192±0.198% | 5.324s |

# D  MINIMALISTIC CODE EXAMPLE FOR ML4TSP SOLVERS

## D.1  TSPSOLVER

To accommodate diverse data types and facilitate the evaluation of solution quality, we provide a **TSPSolver** base class that offers a user-friendly approach for solving TSP. This class includes functionalities for data input and output, as well as an evaluation function. The solver supports different data inputs, such as Numpy arrays and *.txt* and *.tsp* files. The outputs can be saved to corresponding types of files as needed. Additionally, the solver offers an evaluation function, by which users can quickly obtain the average tour length, average gap, and standard deviation of the test dataset. Below is an example of using TSPSolver to obtain the average tour length of the LKH test dataset, as well as its gap with standard deviation relative to the Concorde test dataset.

```python
>>> from ml4tsp import TSPSolver

# Read the test file
>>> solver = TSPSolver()
>>> solver.from_txt("dataset/tsp_uniform/tsp500_lkh.txt")
# Assign the ref_tour of the tested solver as the solution path
>>> solver.read_tours(solver.ref_tours)
# Read the reference file
>>> solver.from_txt("dataset/tsp_uniform/tsp500_concorde.txt")
# (costs_avg, ref_costs_avg, gap_avg, gap_std)
>>> lkh_solver.evaluate(calculate_gap=True)
(16.548110417020276, 16.545805334644392,
0.0139392061675804, 0.028643127914360397)
```

## D.2  LEARNING-BASED SOLVERS

To enhance the modularization and cohesion within our proposed framework, we have extended the **TSPSolver** base class to support two highly integrated learning-based solvers: **ML4TSPNARSolver** and **ML4TSPARSolver**. These solvers can efficiently configure design choices and be called to solve given instances. The following code snippet shows minimalistic code for TSP solving exemplified by solution proximity objective with greedy and MCTS as heuristic search:

```python
>>> from ml4tsp import ML4TSPGNNSolver, ML4TSPNAREnv

# Create ML4TSPEnv
>>> env = ML4TSPNAREnv(
            nodes_num=500, # the number of nodes
            sparse_factor=50 # KNN graph representation
        )

# Create Solver
# If ``pretrained`` is True and ``pretrained_path`` is None,
# ML4TSP will automatically download the corresponding
# pre-trained file online.
>>> solver = ML4TSPGNNSolver(
        env=env,
        encode_device="cuda",
        decoder="random",
        decoder_kwargs={"samples_num": 50},
        local_search="mcts",
```

```
        local_search_kwargs={"time_limit": 1.0},
        local_search_device="cpu",
        scale=1,
        pretrained=True,
        pretrained_path="weights/tsp500_gnn.pt"
    )

# Read data
>>> solver.from_txt("dataset/tsp_uniform/tsp500_test.txt")

# Solve
>>> solver.solve(batch_size=1)

Encoding: 100%|----| 128/128 [00:05<00:00, 25.23it/s]
Encode Time: 9.426303148269653
Decoding: 100%|----| 128/128 [00:00<00:00, 650.48it/s]
Decode Time: 1.578188419342041
Local Search: 100%|----| 6400/6400 [1:57:33<00:00,  1.10s/it]
Local Search Time: 7054.012353181839

# Using the evaluate function to obtain result
>>> solver.evaluate(calculate_gap=True)
(16.59568398521005, 16.545805334644392,
0.3014236256908176, 0.15272430899596431)

# Output and Save the solution results
>>> solver.to_txt("tsp500_gnn_random50_mcts.txt")
```

### D.3 TSPLKHSOLVER AND TSPCONCORDESOLVER

In order to make the traditional C-based solvers, LKH and Concorde, more accessible to Python users, **TSPLKHSolver** and **TSPConcordeSolver** are built upon the TSPSolver framework, providing a user-friendly interface that allows Python users to effectively utilize the powerful capabilities of LKH and Concorde for solving TSP. Here is a simple example solved using the TSPLKHSolver:

```
>>> from ml4tsp import TSPLKHSolver

# 'lkh_max_trials' controls the maximum number of attempts in LKH
>>> lkh_solver = TSPLKHSolver(lkh_max_trials=50000)
>>> lkh_solver.from_txt("data/uniform/test/tsp500.txt")

# Using multi-threading to speed up the solution.
>>> lkh_solver.solve(num_threads=32, show_time=True)
>>> lkh_solver.to_txt("data/uniform/test/tsp500_lkh.txt")
```

### D.4 EVALUATE ON TSPLIB

We filter problems from the TSPLIB dataset with the number of nodes greater than or equal to 51 and less than or equal to 1002, and with EDGE_WEIGHT_TYPE as EUC_2D. We then rescale the node coordinates to the range of 0 to 1 globally. To facilitate future research endeavors on TSP, we use Concorde to precisely solve these problems and produce the corresponding text files. Furthermore,

we provide the **TSPLIB4MLDataset** dataset, which can be used to quickly evaluate TSP solvers. In the experiments of this paper, we selected problems with the number of nodes ranging from 201 to 1002 as the test cases, and below is an illustrative evaluation example.

```python
>>> from ml4tsp import TSPLIB4MLEvaluator, ML4TSPGNNSolver

# Create Evaluator
>>> eval = TSPLIB4MLEvaluator()

# Create Solver
>>> env = ML4TSPNAREnv(sparse_factor=50)
>>> solver = ML4TSPGNNSolver(
        env=env,
        encode_device="cuda",
        decoder="greedy",
        local_search="mcts",
        pretrained_path="weights/tsp500_gnn.pt"
    )

# evaluate
>>> eval.evaluate(solver, min_nodes_num=201, max_nodes_num=1002)
        solved_costs   ref_costs       gaps
ts225      10.713846   10.553828   1.516214
tsp225      7.989420    7.895329   1.191729
pr226       5.290722    5.278835   0.225199
gil262     12.054229   12.049518   0.039099
pr264       6.230572    6.200001   0.493082
a280        9.444607    9.238463   2.231368
pr299       6.730816    6.638419   1.391851
lin318     10.410730   10.169940   2.367660
rd400      15.408393   15.343968   0.419871
fl417       6.424722    6.285529   2.214497
pr439       9.492026    8.990802   5.574861
pcb442     13.469691   13.364091   0.790177
d493        9.474022    9.350242   1.323819
u574       12.127684   12.022998   0.870709
rat575     13.684484   13.619173   0.479546
p654        7.410115    7.195604   2.981133
d657       12.389867   12.211812   1.458056
u724       14.504233   14.436550   0.468836
rat783     15.308325   15.246543   0.405221
pr1002     16.704860   16.396624   1.879873
AVG        10.763168   10.624413   1.416140
```

# E SUPPLEMENTARY EXPERIMENTAL RESULTS AND DISCUSSION

## E.1 FULL RECOMPOSITION RESULTS

We show the full experimental results of learning and searching technique recomposition for ML4TSP solvers in Table 9. Recompositions that exactly correspond to prior methods are noted in the first column.

## E.2 GENERALIZATION TO VARYING DISTRIBUTIONS

Table 4 shows the generalization results of models trained on TSP-500 to varying distributions, e.g., cluster (gaussian mixture), rotation, explosion etc., instead of uniform Zhou et al. (2023). We find that methods with more complex heuristic search methods and methods with online optimization designs may be less affected by distribution shifts. Meanwhile, different methods have their specialization on different data distributions, yet the ranking may not differ much due to their inherent powerfulness.

Table 14: Generalization results of models trained on TSP-500 to varying distributions, e.g., cluster (gaussian mixture), rotation, explosion etc., instead of uniform. The TSP data of different distributions are from Zhou et al. (2023).

| Dataset | Functionality | Training Objective | Learning Search | Construction | Impr. Search | Tour Length | Drop | Time |
|---|---|---|---|---|---|---|---|---|
| TSP300-Cluster_3_10 | Edge Prediction | SL + Node Norm | – | Greedy | MCTS | 9.562 | 1.482% | 0.209s |
| | Edge Pred | SL + Node Norm | – | Random (n=10) | MCTS | 9.493 | 0.732% | 2.016s |
| | Generation | Generative + Symmetry | – | Greedy | Two-Opt | 9.569 | 1.560% | 0.617s |
| | Generation | Generative + Symmetry | Gradient Search | 8 × Greedy | Two-Opt | 9.482 | 0.629% | 10.563s |
| TSP500-Explosion | Edge Pred | SL + Node Norm | – | Greedy | MCTS | 11.911 | 1.298% | 1.245s |
| | Edge Pred | SL + Node Norm | – | Random (n=10) | MCTS | 11.817 | 0.496% | 11.617s |
| | Generation | Generative + Symmetry | – | Greedy | Two-Opt | 11.973 | 1.815% | 0.867s |
| | Generation | Generative + Symmetry | Gradient Search | 8 × Greedy | Two-Opt | 11.853 | 0.807% | 18.021s |
| TSP500-Rotation | Edge Pred | SL + Node Norm | – | Greedy | MCTS | 12.404 | 0.745% | 1.220s |
| | Edge Pred | SL + Node Norm | – | Random (n=10) | MCTS | 12.357 | 0.352% | 11.063s |
| | Generation | SL + Symmetry | – | Greedy | Two-Opt | 12.472 | 1.295% | 0.852s |
| | Generation | SL + Symmetry | Gradient Search | 8 × Greedy | Two-Opt | 12.386 | 0.592% | 17.844s |

## E.3 SUPPLEMENTARY RESULTS FOR FIG. 2

Fig. 3 serves as a supplementary figure to Fig. 2, providing the scatter plots for the correlation between solving performance and the learning loss across reference proximity objectives on TSP-50 test sets.

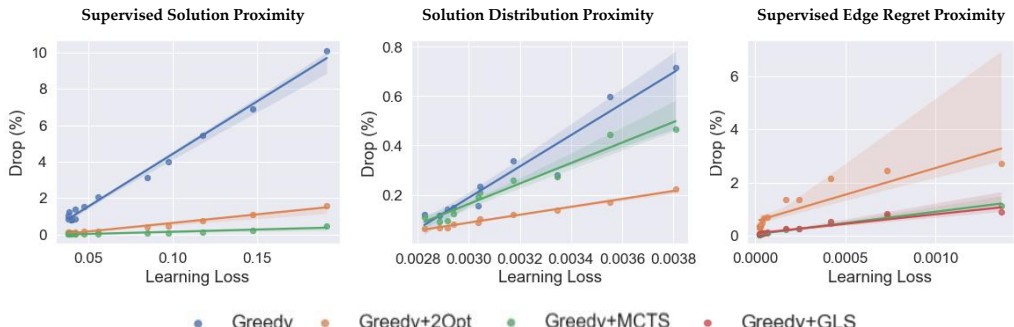

Figure 3: Correlation of performance and learning loss across three reference proximity objectives on TSP50.

## E.4 SUPPLEMENTARY RESULTS OF RUNTIME-DROP CURVE.

Fig. 4 shows the progress of solving within the heuristic search, showing the variation of the drop with respect to search time in the subsequent heuristic search conducted on predicted heatmaps.

## E.5 SUPPLEMENTARY RESULTS FOR SEC. 4.3.4

To further illustrate the positive impact of prediction diversity on the solving performance, based on the solution distribution proximity objective, Fig. 5 shows the effect of increasing the number of sampling in the estimated solution distribution during solving. As can be observed, with the diversity in neural predictions, the solving performance can be improved through a larger sampling number.

## E.6 DISCUSSION OF UNSUPERVISED OBJECTIVES

In edge prediction learning functionality, although unsupervised loss provides seemingly reliable training objectives, we observe that the effectiveness of these methods, including both unsupervised

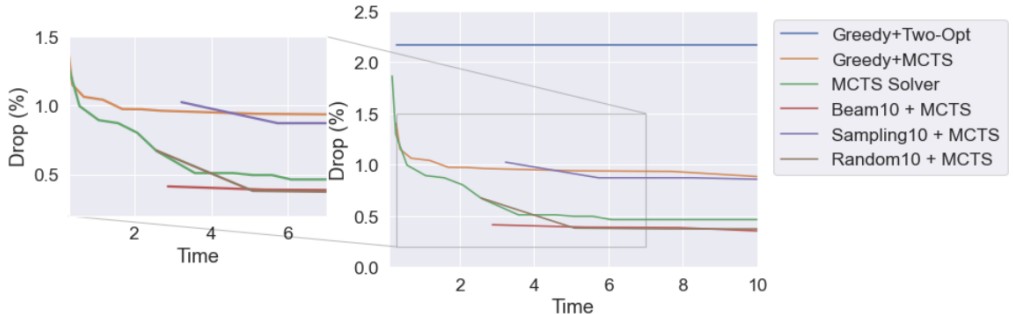

Figure 4: Correlation of performance and runtime.

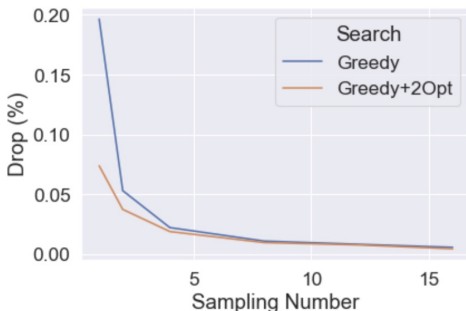

Figure 5: Correlation of performance and sampling number within the estimated distribution for the distribution proximity objective.

solving objective and reinforcement solving objective, relies heavily on heuristic search. Their performance often suffers when subjected to naive construction methods like greedy decoding, with typical drops exceeding 10%. While reproducing these methods, we note that their original implementations heavily relied on the MCTS solver in the search process, where their primary impact lies in providing overall guidance to MCTS. Therefore, incorporating alternative methods during the construction phase to introduce more diversity tends to yield better results. We observe that these methods perform better with Random_MCTS and Beam_MCTS. In particular, we verify that the unsupervised solving objective relies on the Scattering Attention GNN (SAG) (Min et al., 2023) to take effect. Fig. 7 shows the effectiveness of SAG in producing sharper heatmaps, avoiding the over-smooth issue as observed in GCN. However, this SAG-learned heatmap still merely brings a greedy solving performance gain from 20% to 13%. Furthermore, we conducted an in-depth analysis of the correlation between solving performance and learning loss within the framework of unsupervised training, as depicted in Fig. 6. However, it's disappointed to find that a decrease in learning loss did not result in an improvement in solving performance. Thus, it is crucial for edge prediction based unsupervised designs to develop more compatible heuristic search and pursue better integration.

### E.7   DISCUSSION OF SUPERVISION QUALITY

We observe that the default setting of LKH3 (Helsgaun, 2017) does not generally produce the optimal solutions as claimed by previous works (Qiu et al., 2022; Sun & Yang, 2023). We experiment by adjusting the *max_trials* hyperparameter in LKH. We gradually increase it and observe that, for TSP-500, the solution quality continues to improve even as we increase the number of trials from 10,000 to 50,000. In this paper, we utilize LKH3 as our source for supervised signals. We employ 500, 5000, and 50000 trials for TSP-50, TSP-100, and TSP-500, respectively.

### E.8   DISCUSSION OF MCTS

We make an intriguing observation investigating MCTS that the tour length before and after applying MCTS does not show a straightforward proportional relationship. This observation serves as the

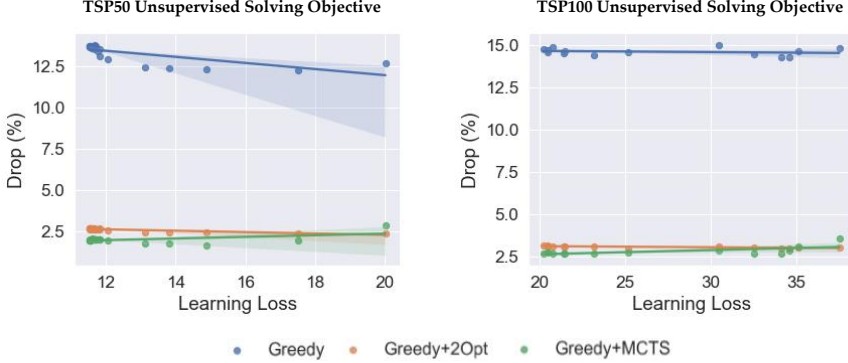

Figure 6: Correlation of performance and learning loss on the unsupervised solving objective.

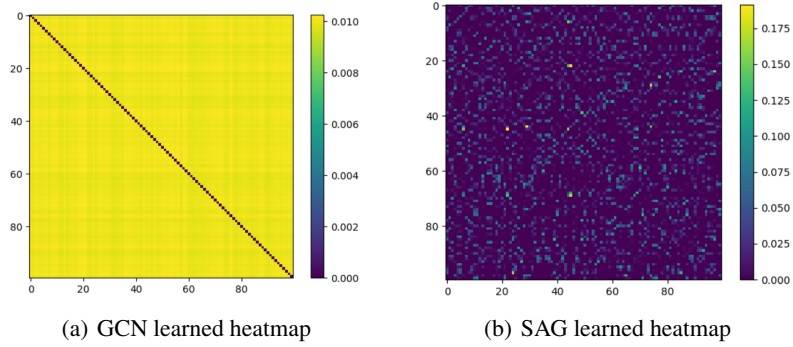

(a) GCN learned heatmap          (b) SAG learned heatmap

Figure 7: Edge heatmap prediction learned by different graph networks.

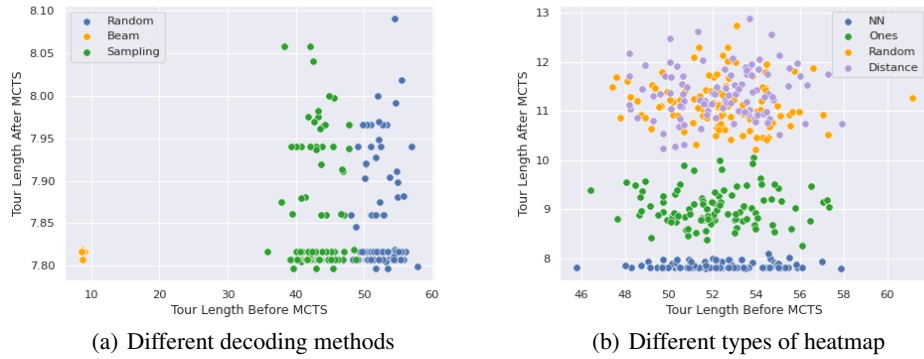

(a) Different decoding methods          (b) Different types of heatmap

Figure 8: Effects of different decoding methods and heatmap types on MCTS

motivation for us to investigate and develop a range of search methods rooted in MCTS, each incorporating distinct decoding strategies: *Random_MCTS*, *Beam_MCTS*, and *Sampling_MCTS*.

Fig. 8 (a) illustrates the variation in tour length before and after applying these different search methods. The experiments are conducted on the same TSP-100 instance utilizing the same heatmap generated by the model trained with solution proximity objective with node-wise normalization. Each method is sampled 100 times. Note *Beam_MCTS*, employing beam search as the sampling strategy, achieves notably shorter average tour lengths before the heuristic search compared to the other two strategies. However, its performance after applying MCTS is found to be comparatively inferior

to the other two strategies, as we consistently select the best result obtained from all the samples for each method. This finding aligns with the results presented in Table 9, where, with a sampling number of 50, *Beam_MCTS* exhibits an average drop of 0.013%, while the other two methods show a drop of 0.003%, 0.002% respectively.

Furthermore, we investigate the impact of the predicted heatmaps on MCTS. Four types of heatmaps are examined: heatmaps generated by neural networks (NN), all-one heatmaps (Ones), random heatmaps (Random), and distance heatmaps (Distance). Fig. 8 (b) illustrates that the heatmap obtained from the neural network achieved the highest performance compared to the other types. This finding further confirms the significant role of machine learning in enhancing the MCTS method to solve TSP problems.

# F    SUPPLEMENTARY DETAILS OF INCORPORATED METHODS

## F.1    TRAINING OBJECTIVES

**Supervised Solution Proximity.** In implementation, we adopt GCN for edge prediction, which receives the node coordinates as the input features and the output edge features would be further mapped by an MLP to predict the probability of including this edge in the optimal solution. The weights of the weighted binary cross-entropy loss are given as $w_0 = \frac{n^2}{(n^2-2n)\times c}$ and $w_1 = \frac{n^2}{(2n)\times c}$ where $c = 2$ denotes the number of classes.

**Supervised Edge Regret Proximity.** Using the same GCN+MLP network architecture, the model is trained with mean squared error loss to predict the regret associated with each edge. Due to the high computational complexity, obtaining the optimal solutions for calculating the corresponding regret is not tractable. Therefore, supervision signals are obtained through LKH3 (Helsgaun, 2017). The max_trials parameters for TSP-50 and 100 are set as 10 and 5, respectively. In this process, a global reference solution is computed, and the supervision for approximate regret is derived by fixing a particular edge and assessing the degradation in the reference solution caused by fixing that edge.

**Solution Distribution Proximity.** In this paper, we mainly implement the discrete diffusion models (Sun & Yang, 2023; Li et al., 2023b) as the representative of generative model based solvers. Taking discrete diffusion models (Sun & Yang, 2023; Li et al., 2023b) as representation in this paper, which generates solutions $\mathbf{S}_0 \in \{0,1\}^{n \times n}$ by $T$-step denoising process from random noises $\mathbf{S}_T$, and the latent variables include noised solution $\mathbf{S}_{1:T}$. For each entry, the model estimates a Bernoulli distribution indicating whether this entry should be selected. In implementation, each entry of solution is represented by a one-hot vector[3] such that $\mathbf{S}_0 \in \{0,1\}^{n \times n \times 2}$. Following the notations of Ho et al. (2020); Austin et al. (2021), the general framework of diffusion includes a forward noising and a reverse denoising Markov process. The noising process takes the initial solution $\mathbf{S}_0$ and progressively introduces noise to generate a sequence of latent variables $\mathbf{S}_{1:T}$. The denoising process is learned by the model, which starts from the final latent variable $\mathbf{S}_T$ and denoises $\mathbf{S}_t$ at each time step to generate the preceding variables $\mathbf{S}_{t-1}$ based on the instance $G$, eventually recovering the target data distribution. The formulation of the denoising process is expressed as $p_\theta(\mathbf{S}_{0:T}|G) = p(\mathbf{S}_T)\prod_{t=1}^{T} p_\theta(\mathbf{S}_{t-1}|\mathbf{S}_t, G)$. The training optimization aims to align $p_\theta(\mathbf{S}_0|G)$ with the data distribution $q(\mathbf{S}_0|G)$ using ELBO:

$$\mathcal{L} = \mathbb{E}_q\left[ \sum_{t>1} D_{KL}\left[ q(\mathbf{S}_{t-1}|\mathbf{S}_t, \mathbf{S}_0) \parallel p_\theta(\mathbf{S}_{t-1}|\mathbf{S}_t, G)\right] - \log p_\theta(\mathbf{S}_0|\mathbf{S}_1, G)\right] + C \tag{6}$$

Specifically, the forward noising process is achieved by multiplying $\mathbf{S}_t \in [0,1]^{N \times N \times 2}$ at step $t$ with a forward transition probability matrix $\mathbf{Q}_t \in [0,1]^{2 \times 2}$ where $[\mathbf{Q}_t]_{i,j}$ indicates the probability of transforming $E_i$ in each entry to $E_j$. We set $\mathbf{Q}_t = \begin{bmatrix} \beta_t & 1-\beta_t \\ 1-\beta_t & \beta_t \end{bmatrix}$ (Austin et al., 2021), where $\beta_t \in [0,1]$ such that the transition matrix is doubly stochastic with strictly positive entries, ensuring that the stationary distribution is uniform which is an unbiased prior for sampling. The noising process for each step and the $t$-step marginal are formulated as:

$$q(\mathbf{S}_t|\mathbf{S}_{t-1}) = \text{Cat}(\mathbf{S}_t; \mathbf{p} = \mathbf{S}_{t-1}\mathbf{Q}_t) \quad \text{and} \quad q(\mathbf{S}_t|\mathbf{S}_0) = \text{Cat}(\mathbf{S}_t; \mathbf{p} = \mathbf{S}_0\overline{\mathbf{Q}}_t) \tag{7}$$

---

[3]Each entry with $[0,1]$ indicates that it is included in $\mathbf{S}$ and $[1,0]$ indicates the opposite.

where $\text{Cat}(\mathbf{S}; \mathbf{p})$ is a categorical distribution over $N$ one-hot variables with probabilities given by vector $\mathbf{p}$ and $\overline{\mathbf{Q}}_t = \mathbf{Q}_1 \mathbf{Q}_2 \cdots \mathbf{Q}_t$. Through Bayes' theorem, the posterior can be achieved as:

$$q(\mathbf{S}_{t-1}|\mathbf{S}_t, \mathbf{S}_0) = \frac{q(\mathbf{S}_t|\mathbf{S}_{t-1}, \mathbf{S}_0) q(\mathbf{S}_{t-1}|\mathbf{S}_0)}{q(\mathbf{S}_t|\mathbf{S}_0)} = \text{Cat}\left(\mathbf{S}_{t-1}; \mathbf{p} = \frac{\mathbf{S}_t \mathbf{Q}_t^\top \odot \mathbf{S}_0 \overline{\mathbf{Q}}_{t-1}}{\mathbf{S}_0 \overline{\mathbf{Q}}_t \mathbf{S}_t^\top}\right) \quad (8)$$

The neural network is trained to predict the logits of the distribution $\tilde{p}_\theta(\tilde{\mathbf{S}}_0|\mathbf{S}_t, G)$, such that the denoising process can be parameterized through $q(\mathbf{S}_{t-1}|\mathbf{S}_t, \tilde{\mathbf{S}}_0)$:

$$p_\theta(\mathbf{S}_{t-1}|\mathbf{S}_t) \propto \sum_{\tilde{\mathbf{S}}_0} q(\mathbf{S}_{t-1}|\mathbf{S}_t, \tilde{\mathbf{S}}_0) \tilde{p}_\theta(\tilde{\mathbf{S}}_0|\mathbf{S}_t, G) \quad (9)$$

**Reinforcement Solving Objective.** For edge prediction functionality, to calculate the gradient estimation by REINFORCE as presented in Eq. 4, we follow Qiu et al. (2022) to build the auxiliary distribution for TSP. Representing the solution as a permutation $\pi$, the auxiliary distribution can be formulated as:

$$q_\theta(\pi|G) := \sum_{j=0}^{n-1} \frac{1}{n} \cdot q_\theta(\pi|\pi_0 = j, G) = \sum_{j=0}^{n-1} \frac{1}{n} \cdot \prod_{i=1}^{n-1} q_\theta(\pi_i|\pi_{<i}, G) = \sum_{j=0}^{n-1} \frac{1}{n} \cdot \prod_{i=1}^{n-1} \frac{\exp(\hat{\mathbf{H}}_{\pi_{t-1}, \pi_t})}{\sum_{j=i}^n \exp(\hat{\mathbf{H}}_{\pi_{t-1}, \pi_t})} \quad (10)$$

where $\hat{\mathbf{H}}$ represents the estimated edge heatmap. With this auxiliary distribution, in each REINFORCE update, we sample 2,000 samples for TSP-50/100 and 120 samples for TSP-500 from $q_\theta(\pi|G)$ to estimate the gradients.

For the learning functionality of sequential node prediction, instead of establishing the per-step node heatmap from the estimated edge heatmap, sequence models decode each step's node heatmap sequentially. The attention based encoder-decoder model defines a stochastic policy $p(\pi|G)$ by modeling $p_\theta(\pi_t|\pi_{t-1}, G)$ at each step. The encoder produces embeddings of all the nodes and the decoder takes the node embeddings, and context of the current state as input, then it uses the attention mechanism to output the attention map over the nodes. With visited nodes masked, the attention map over other unvisited nodes serves as the per-step node heatmap. The gradients are estimated by Eq. 4, where the auxiliary distribution is established upon node heatmaps.

The symmetric baseline on top of the sequential node prediction objective optimized with RL is implemented referring to Kim et al. (2022). The training objective leverages the symmetricities of CO by incorporating the regularization loss of invariant representation $\mathcal{L}_{inv}$, the REINFORCE loss with solution symmetricity regularization $\mathcal{L}_{ss}$, and the REINFORCE loss with both solution and problem symmetricity regularization $\mathcal{L}_{ps}$. The final loss is written as $\mathcal{L}_{total} = \mathcal{L}_{ps} + \alpha \mathcal{L}_{inv} + \beta \mathcal{L}_{ss}$. $\mathcal{L}_{ss}$ regularizes the symmetric solutions to have the same objective values:

$$\mathcal{L}_{ss} = -\mathbb{E}_{p_\theta(\pi|G)}[R(\pi; G)]$$
$$\nabla \mathcal{L}_{ss} \approx -\frac{1}{K} \sum_{k=1}^K \left[ [R(\pi^k; G) - \frac{1}{K} \sum_{k=1}^K R(\pi_k; G)] \nabla_\theta \log p_\theta \right] \quad (11)$$

where $\{\pi^k\}_{k=1}^K$ are the solutions from $p_\theta(\pi|G)$. $\mathcal{L}_{ps}$ regularizes the problem symmetricity when two problems have identical optimal solution sets, which is obtained through rotation:

$$\mathcal{L}_{ps} = -\mathbb{E}_{Q^l \sim \mathcal{Q}} \mathbb{E}_{p_\theta(\pi|Q^l(G))}[R(\pi; G)]$$
$$\nabla \mathcal{L}_{ss} \approx -\frac{1}{LK} \sum_{l=1}^L \sum_{k=1}^K \left[ [R(\pi^{l,k}; G) - \frac{1}{LK} \sum_{l=1}^L \sum_{k=1}^K R(\pi_{l,k}; G)] \nabla_\theta \log p_\theta \right] \quad (12)$$

where $\mathcal{Q}$ is the distribution of random orthogonal matrices, $Q^l$ is the $l^{th}$ sampled rotation matrix, and $\pi^{l,k}$ is the $k^{th}$ sample solution of the $l^{th}$ rotated problem. $\mathcal{L}_{inv}$ regularizes the model to predict close hidden representations of symmetric problems:

$$\mathcal{L}_{inv} = -S_{cos}\left(\text{MLP}\left(h_\theta(G)\right), \text{MLP}\left(h_\theta\left(Q(G)\right)\right)\right) \quad (13)$$

where $S_{cos}$ is the cosine similarity metric, $h_\theta$ is the neural feature projection, $Q(G)$ is the rotated problem, and MLP is a projection head.

## F.2 LEARNING-BASED SEARCH

**Gradient Search.** Following T2T (Li et al., 2023b), the algorithm starts with an initial solution $\mathbf{S}_0$ and conducts several iterations to enhance the given solution. Each iteration involves adding a certain degree of noise to disrupt the structure, denoising with objective gradient guidance to obtain a lower-cost solution. The algorithm eventually reports the solution with the lowest objective score. To introduce a controlled degree of disruption to the given solution, we employ $\mathbf{S}_0\overline{\mathbf{Q}}_{\alpha T}$ to derive the distribution of the disrupted solution $q(\mathbf{S}_{\alpha T}|\mathbf{S}_0)$ as $N$ Bernoulli distributions, where $\alpha$ serves as a hyperparameter to control the degree of noise. Subsequently, the disrupted solution $\mathbf{S}_{\alpha T}$ can be sampled from $q(\mathbf{S}_{\alpha T}|\mathbf{S}_0)$. From $\mathbf{S}_{\alpha T}$, we employ $p_\theta(\mathbf{S}_{t-1}|\mathbf{S}_t, G, y^*)$ to perform denoising, which aimed at recovering a potentially lower-cost $\mathbf{S}_0'$.

## F.3 LOCAL SEARCH

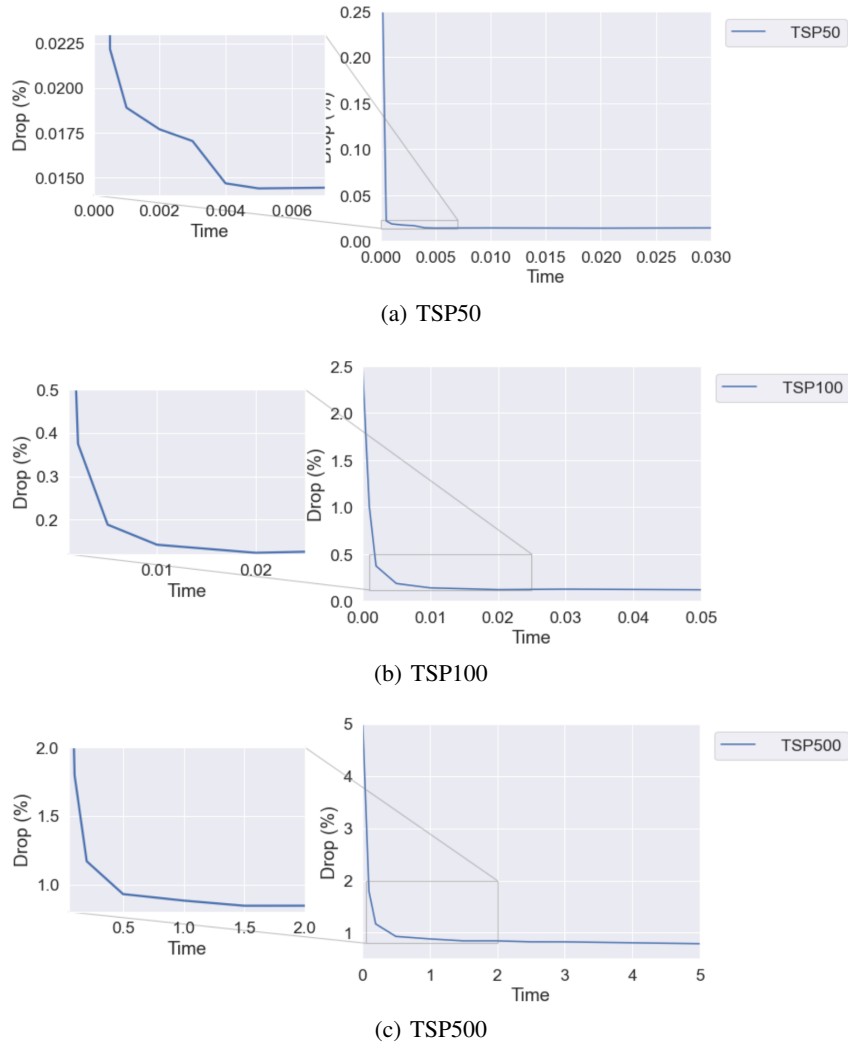

(a) TSP50

(b) TSP100

(c) TSP500

Figure 9: The relationship between the solution quality and the search time of MCTS for different problem scales.

**Two-Opt.** For a TSP tour sequence $x_1, ..., x_p, x_{p+1}, ..., x_q, x_{q+1}, ...$, suppose we select two nodes $x_p$ and $x_q$, and perform a Twp-Opt operation by swapping the subsequence between them, the resulting path becomes $x_1, ..., x_p, x_q, x_{q-1}..., x_{p+1}, x_{q+1}$. The difference between the path lengths before and after the swap is denoted as $reward = D_{p,p+1} + D_{q,q+1} - D_{p,q} - D_{p+1,q+1}$, where $D_{i,j}$ represents

the distance between node $i$ and node $j$. In each iteration, the two nodes that can obtain the maximum $reward$ are selected to perform the Twp-Opt operation. The iteration terminates when either no more improvements or the maximum number of iterations has been reached. In the experiments of this paper, the maximum number of iterations was set to 5,000.

**Monte Carlo Tree Search.** In our MCTS implementation, we adopt the framework of the MCTS Solver (Fu et al., 2021) while refining certain design aspects.

The first improvement is the Two-Opt module. The MCTS Solver's Two-Opt strategy involves performing the swap immediately upon identifying a pair of nodes that can yield a gain. In contrast, our approach aligns with the raw Two-Opt implementation, where we select the optimal pair of nodes in each iteration.

The second aspect pertains to our observation that the MCTS derived from the MCTS Solver underutilizes the allocated time slice. When a better solution cannot be found with the current random seed, the search terminates prematurely. Additionally, we discover that continuously replacing the random seed somewhat guides the search toward improved solutions, especially on large-scale problems like TSP500, where the gap of the model, which trains with solution proximity objective with node-wise normalization, is reduced from 0.900% to 0.578%.

Based on observation and improvement, we study the optimal search time for MCTS under different problem scales. We take the heatmap encoded by the GNN model and the tour obtained by greedy decoding of this heatmap as a baseline for this investigation. The results are presented in Fig. 9. Based on the results, we set the time limits for the MCTS on problems with 50, 100, and 500 nodes as 0.005, 0.020, and 1.000 seconds, respectively.

Note that although continuous search offers benefits for small-scale problems, the gains are marginal when weighed against the extra time spent. Therefore, to improve the solving speed, we retain the original fixed random number scheme for the TSP50 and TSP100. However, for the TSP500 problem, we implement the optimized scheme to enhance efficiency.

# G  NETWORK ARCHITECTURE DETAILS

The network backbones include GNN variants like Graph Convolutional Networks (GCN) (Kipf & Welling, 2016; Joshi et al., 2019), Graph Attention Networks (GAT) (Veličković et al., 2017; Kool et al., 2018), and Scattering Attention GNN (SAG) (Min et al., 2023). In the implementation of non-autoregressive methods, we employ GCN as the model backbone, while for autoregressive methods, we use the transformer (Kool et al., 2018) architecture to build the sequence model, with GAT serving as the essential network backbone. We ensure consistency in GCN and GAT implementations, including the number of layers and feature dimensions. In particular, for the unsupervised solving objectives for edge prediction, due to their reliance on the signal filtering capability of the SAG network, we utilize the SAG network in the implementation of this method, while again keeping its number of layers and feature dimensions consistent.

## G.1  GRAPH CONVOLUTIONAL NETWORKS

**Input Embedding Layer.** Given node vector $x \in \mathbb{R}^{N \times 2}$ and weighted edge vector $e \in \mathbb{R}^{E}$, we compute the sinusoidal features of each input element respectively. The denoising timestep $t$ is optional, and it is required when using the diffusion model. Here, $N$ denotes the number of nodes in the graph, and $E$ denotes the number of edges, $t \in \tau_1, \ldots, \tau_M$.

$$\tilde{x}_i = \mathrm{concat}(\tilde{x}_{i,0}, \tilde{x}_{i,1}) \tag{14}$$

$$\tilde{x}_{i,j} = \mathrm{concat}\left(\sin \frac{x_{i,j}}{T^{\frac{0}{d}}}, \cos \frac{x_{i,j}}{T^{\frac{0}{d}}}, \sin \frac{x_{i,j}}{T^{\frac{2}{d}}}, \cos \frac{x_{i,j}}{T^{\frac{2}{d}}}, \ldots, \sin \frac{x_{i,j}}{T^{\frac{d}{d}}}, \cos \frac{x_{i,j}}{T^{\frac{d}{d}}}\right) \tag{15}$$

$$\tilde{e}_i = \mathrm{concat}\left(\sin \frac{e_i}{T^{\frac{0}{d}}}, \cos \frac{e_i}{T^{\frac{0}{d}}}, \sin \frac{e_i}{T^{\frac{2}{d}}}, \cos \frac{e_i}{T^{\frac{2}{d}}}, \ldots, \sin \frac{e_i}{T^{\frac{d}{d}}}, \cos \frac{e_i}{T^{\frac{d}{d}}}\right) \tag{16}$$

$$\tilde{t} = \mathrm{concat}\left(\sin \frac{t}{T^{\frac{0}{d}}}, \cos \frac{t}{T^{\frac{0}{d}}}, \sin \frac{t}{T^{\frac{2}{d}}}, \cos \frac{t}{T^{\frac{2}{d}}}, \ldots, \sin \frac{t}{T^{\frac{d}{d}}}, \cos \frac{t}{T^{\frac{d}{d}}}\right) \tag{17}$$

where $d$ is the embedding dimension, $T$ is a large number (usually selected as $10000$), $\text{concat}(\cdot)$ denotes concatenation.

Next, we compute the input features of the graph convolution layer:

$$x_i^0 = W_1^0 \tilde{x}_i \tag{18}$$

$$e_i^0 = W_2^0 \tilde{e}_i \tag{19}$$

$$t^0 = W_4^0(\text{ReLU}(W_3^0 \tilde{t})) \tag{20}$$

where $t^0 \in \mathbb{R}^{d_t}$, $d_t$ is the time feature embedding dimension. Specifically, for TSP, the embedding input edge vector $e$ is a weighted adjacency matrix, which represents the distance between different nodes, and $e^0$ is computed as above.

**Graph Convolution Layer.** Following Joshi et al. (2019), the cross-layer convolution operation is formulated as:

$$x_i^{l+1} = x_i^l + \text{ReLU}(\text{BN}(W_1^l x_i^l + \sum_{j \sim i} \eta_{ij}^l \odot W_2^l x_j^l)) \tag{21}$$

$$e_{ij}^{l+1} = e_i^l + \text{ReLU}(\text{BN}(W_3^l e_{ij}^l + W_4^l x_i^l + W_5^l x_j^l)) \tag{22}$$

$$\eta_{ij}^l = \frac{\sigma(e_{ij}^l)}{\sum_{j' \sim i} \sigma(e_{ij'}^l) + \epsilon} \tag{23}$$

where $x_i^l$ and $e_{ij}^l$ denote the node feature vector and edge feature vector at layer $l$, $W_1, \cdots, W_5 \in \mathbb{R}^{h \times h}$ denote the model weights, $\eta_{ij}^l$ denotes the dense attention map. The convolution operation integrates the edge feature to accommodate the significance of edges in routing problems.

For TSP, we aggregate the timestep feature with the edge convolutional feature and reformulate the update for edge features as follows:

$$e_{ij}^{l+1} = e_{ij}^l + \text{ReLU}(\text{BN}(W_3^l e_{ij}^l + W_4^l x_i^l + W_5^l x_j^l)) + W_6^l(\text{ReLU}(t^0)) \tag{24}$$

**Output Layer.** The prediction of the edge heatmap in TSP is as follows:

$$e_{i,j} = \text{Softmax}(\text{norm}(\text{ReLU}(W_e e_{i,j}^L))) \tag{25}$$

where $L$ is the number of GCN layers and norm is layer normalization.

## G.2 GRAPH ATTENTION NETWORKS

**Input Embedding Layer.** Given node vector $x \in \mathbb{R}^{N \times 2}$, we compute initial $d_h$-dimensional node embeddings $h_i^{(0)}$ through a learned linear projection with parameters $W^x$ and $b^x$: $h_i^{(0)} = W^x x_i + b^x$

**Attention Layer.** Following Veličković et al. (2017); Kool et al. (2018), we denote with $h_1^{(l)}$ the node embeddings produced by layer $l \in \{1, ..., N\}$ and the attention layer operation is formulated as:

$$\hat{h}_i = \text{BN}^l(h_i^{(l-1)} + \text{MHA}_i^l(h_1^{(l-1)}, ..., h_n^{(l-1)})) \tag{26}$$

$$h_i^{(l)} = \text{BN}^l(\hat{h}_i + \text{FF}^l \hat{h}_i) \tag{27}$$

where BN represents batch normalization, MHA denotes the multi-head attention layer, and FF refers to the node-wise fully connected feed-forward layer.

## G.3 SCATTERING ATTENTION GNN

**Input Embedding Layer.** Given node vector $x \in \mathbb{R}^{N \times 2}$, we compute initial $d_h$-dimensional node embeddings $H^0$ through a learned linear projection with parameters $W^x$ and $b^x$: $H^0 = W^x x_i + b^x$

**Diffusion Module.** Following Min et al. (2023), the diffusion module consists of a cascade of $K \in \mathbb{N}$ aggregation (or diffusion) layers with operations that are chosen for each node via an attention

Table 15: Design choices of existing major methods.

| Method | Offline Learning | | Offline Solving | | Paradigm |
|---|---|---|---|---|---|
| | Learning Functionality | Training Objective | Learning-based Search | Heuristic Search | |
| GCN (Joshi et al., 2019) | Edge Prediction | Supervised Solution Proximity | None | Greedy / Beam Search | b) c) |
| Att-GCN (Fu et al., 2021) | Edge Prediction | Supervised Solution Proximity | None | MCTS | c) |
| DIMES (Qiu et al., 2022) | Edge Prediction | (Meta) Reinforcement Solving Objective | Active Search | Greedy / Sampling / MCTS | b) c) |
| UTSP (Min et al., 2023) | Edge Prediction | Unsupervised Solving Objective | None | Best-first Local Search | c) |
| GNNGLS (Hudson et al., 2022) | Edge Regret Prediction | Supervised Edge Regret Proximity | None | Guided Local Search | c) |
| DIFUSCO (Sun & Yang, 2023) | Solution Generation | Solution Distribution Proximity | None | Greedy / Sampling / MCTS | b) c) |
| T2T (Li et al., 2023b) | Solution Generation | Solution Distribution Proximity | Gradient Search | Greedy / Sampling / MCTS | c) |
| AM (Kool et al., 2018) | Sequential Node Prediction | Reinforcement Solving Objective | None | Greedy / Sampling | b) |
| POMO (Kwon et al., 2020) | Sequential Node Prediction | Reinforcement Solving Objective | None | Greedy / Sampling | b) |
| EAS (Hottung et al., 2021b) | Sequential Node Prediction | Reinforcement Solving Objective | Active Search | Greedy | b) |
| Sym-NCO (Kim et al., 2022) | Sequential Node Prediction | Reinforcement Solving Objective | None | Greedy / Sampling | b) |

mechanism. In each layer $l$, every node has access to node representations from a set of filters $\mathcal{F}$ that contains a selection of low-pass and band-pass filters as follows:

$$f_{low,r}(H^{l-1}) = A^r H^{l-1} \tag{28}$$

$$f_{band,k}(H^{l-1}) = \Psi_k H^{l-1} \tag{29}$$

where $\Psi_k \in \mathbb{R}^{n \times n}$ represents a wavelet (Wenkel et al., 2022) at scale $2^k$.

To assess the relevance of filter $f$ to node $v$, data-driven scores $s_f(v)$ are computed. The variable $H^l_f$ represents the outcome of applying filter $f$ to $H^{l-1}$. These scores are determined using an attention mechanism, which can be defined as:

$$s^l_f := \sigma(H^l_f || H^{l-1}) \, \mathbf{a}^l \tag{30}$$

where $||$ denotes the concatenation operation and $\mathbf{a}^l \in \mathbb{R}^{2d_h}$ denotes the attention vector.

Moreover, the scores are normalized across the filters using the softmax function, denoted as $\alpha_f(v) = \text{Softmax}_{\mathcal{F}}(s_f(v))$. These normalized scores are stored in $\alpha^l_f$. Next, the node representations are updated through the following process:

$$H^l_{agg} := \sum_{f \in \mathcal{F}} \alpha^l_f \odot H^l_f \tag{31}$$

where $\odot$ is the element-wise multiplication (applied separately to each column of $H^l_f$)

**Output Layer.** The formulas for the output layer is as follows:

$$T = \text{Softmax}(m^l(H^l_{agg})) \tag{32}$$

where $m^l$ is the MLP: $\mathbb{R}^{d_h} \to \mathbb{R}^{d_h}$. where $m^l$ is the MLP: $\mathbb{R}^{d_h} \to \mathbb{R}^{d_h}$. For the TSP problem, $T$ represents the assignment matrix in the output, and subsequently, the edge heatmap will be obtained by $T V T^T$.

# H    DISCUSSION FOR THE SYNERGY OF TRAINING AND SOLVING FOR CO

Recall that the solving process of traditional CO solvers primarily involves an online solving procedure based on manually designed heuristics, while the introduction of learning techniques has recently brought about a paradigm shift in this field. In this section, we present the general formulation of CO and specifically TSP, as well as three paradigms for how existing CO solvers exploit machine learning, which serves as a design guideline for our modular framework.

**Pure Online Solving.** This paradigm, as shown in Fig. 10 (a), typically comprises classic exact (Applegate et al., 2006) and heuristic (Lin & Kernighan, 1973; Helsgaun, 2017; Fu et al., 2021; Silver et al., 2016) solvers, which operate in a real-time manner without any prior offline learning phase. These solvers rely solely on the data received on the fly according to some unknown distribution and do not leverage any pre-learned knowledge or historical data. They make immediate decisions based on the features of the current problem instance and the objective is to minimize the objective cost for every encountered instance. This classic paradigm has been well-optimized and verified by the operation research community.

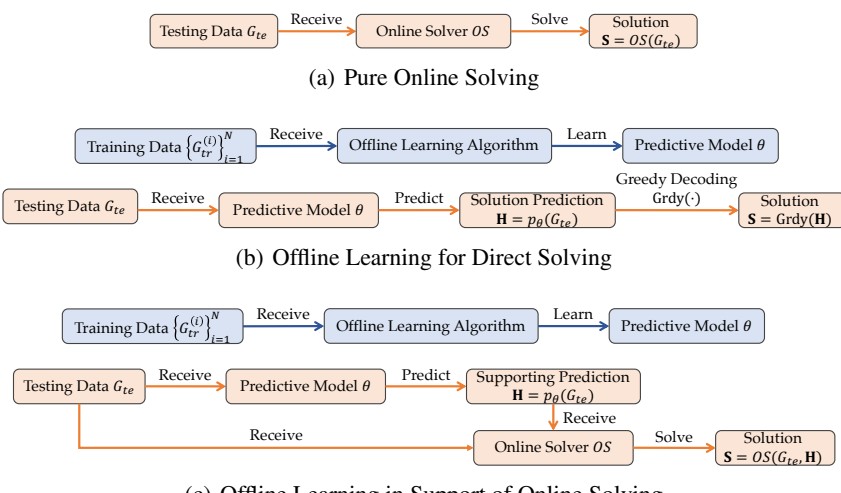

Figure 10: Solving paradigms for existing mainstream CO solvers. Blue and orange stand for offline and online processes, respectively.

However, instances in real-world applications are often from a specific distribution with similar patterns and characteristics (Bengio et al., 2021), e.g., the scheduling of daily deliveries within a particular region may exhibit commonalities. Since this paradigm learns nothing from the pre-existing data, the computation for each instance can be inefficiently repetitive when the data exhibits structural similarity.

**Offline Learning for Direct Solving.** In this paradigm as shown in Fig. 10 (b), the neural solver undergoes an offline learning phase where it learns from historical data before engaging in online solving. The offline learning algorithms produce predictive models that can directly predict solutions based on the given instance features. The primary goal is to learn the predictive models that produce solution predictions with the minimized average estimation error (for supervised learning) or average objective cost (for unsupervised learning) compared to the optimal solutions.

However, following the classic machine learning mechanism, the neural network is optimized by minimizing the average objective score across the distribution of historical problem instances, diverging from the core target of CO of seeking optimal solutions for every newly encountered instance (Li et al., 2023b; Wang & Li, 2022). Without a particular procedure of online search for the specific instance, the predictive model struggles to adapt quality solutions to newly encountered instances, and the typical local minima convergence limits further active search.

**Offline Learning in Support of Online Solving.** This paradigm, as shown in Fig. 10 (c), incorporates an offline learning phase that complements the online solving process. During the offline phase, the solver gathers insights from historical data, which are then leveraged to bolster and optimize the efficiency of the online problem-solving procedures. The offline learning phase can provide valuable support to the online solver through various means, including solution initialization for online solving e.g. naive network prediction (Joshi et al., 2019; Fu et al., 2021), solution distribution estimation e.g. generative modeling (Sun & Yang, 2023; Li et al., 2023b), and clues to searching for higher quality solutions e.g. regret estimation for local search (Hudson et al., 2022).

**Overview.** In general, offline learning as a data-driven process can serve as a complement to the per-instance online solving process, such that the solver can excel in both efficiency in processing instances with similar structure and generalizing to every newly encountered instance with sufficient specialized search. Table 15 summarizes the current mainstream algorithms for the TSP problem and categorizes them into the above paradigms to facilitate the follow-up detailed analysis.

# I  EXPERIMENTAL DETAILS

All the experiments are performed on GPUs of NVIDIA RTX3090 and a CPU of AMD Ryzen Threadripper 3970X 32-Core Processor. All the test evaluations are performed on a single GPU.

## I.1  IMPLEMENTATION AND HYPERPARAMETERS

Graph convolutional networks:

- **Feature dimensions**: the dimension of the input features for implemented graph networks equals 2 indicating the 2D coordinates of the nodes; the feature dimension of the intermediate layers is set as 256; The output channel dimensions of the networks are set as 2 for TSPGNN, TSPGNNWISE, and TSPDiffusion, and 1 for TSPDIMES , TSPGNN4REG and USTP.

- **Number of layers**: default be set as 12.

- **Sparse factor**: for TSP500, in order to reduce computational and GPU memory requirements, the sparse factor is set as 50, which indicates that each node in the network only considers its nearest 50 nodes for computations. While for TSP50 and TSP100, sparse processing is typically not employed, hence their sparse factor is set as -1.

Graph attention networks:

- **Feature dimensions**: the feature dimension of the intermediate layers is set as 128.

- **Number of layers**: default be set as 3.

- **Number of heads:** the number of heads for the attention is set as 8.

Scattering attention GNN:

- **Feature dimensions**: the feature dimension of the intermediate layers is set as 64.

- **Number of layers**: default be set as 3.

Training objectives:

- **Solution distribution proximity:** We implement the model with 50 inference steps for denoising, and the models are trained with 1000 denoising steps, i.e., $T = 1000$. We additionally follow Sun & Yang (2023); Li et al. (2023b) to apply the technique of denoising diffusion implicit models (DDIMs) (Song et al., 2020) for accelerating inference and solution reconstruction.

- **Meta learning objective:** The number of epochs for the inner optimization loop is set as 100, and the learning rate is 5e-2. The outer loop can be exactly viewed as the process of active search.

- **Unsupervised solving objective:** The weights of the Hamiltonian cycle losses and the no self-loop constraints, i.e. $\lambda_1$ and $\lambda_2$ in Eq. 3, are set as 10 and 0.1, respectively.

Learning-based search methods:

- **Gradient Search**: ratio of noise adding in rewrite steps equals 0.4; the number of rewriting steps i.e. guided denoising steps is set as 5 for TSP-50 and 100, and 10 for TSP-500.

- **Active Search**: as_steps is set as 100 and as_samples is set as 1000.

Improved search methods:

- **Two-opt Search**: The maximum number of iterations is set as 5000.

- **Guided LS**: The maximum number of moves is set as 100, and the maximum search time is limited to 10 seconds.

- **MCTS**: The maximum search time is one-tenth of the total number of nodes, and the maximum search depth is 10.

## I.2 DETIALS OF EXPERIMENTS

**Training Details.** All models are trained with a cosine learning rate schedule starting from $2 \times 10^{-4}$ and ending at $0$. For TSP-50, we generally use 1,280,000 random instances for training and train the models for 100 epochs. For TSP-100, we use 1,280,000 random instances and train the models for 50 epochs. We apply curriculum learning and initialize the models from TSP-50 checkpoints. For TSP-500, we use 128,000 random instances and train the models for 50 epochs. We also apply curriculum learning and initialize the models from TSP-100 checkpoints.

**Experimental Details of Fig. 2.** To reflect the correlation between the solving performance and the offline learning quality, we show the scatter plots for the optimality drops with the corresponding learning objective values. During training, we equivalently sample 10 time points for recording and additionally sample the first 5 epochs for their importance as key representatives in the trends. The solid straight lines in the figures represent the fitted trend based on the scatters.

**Experimental Details of Fig. 4.** As the improvement-based search continuously optimizes within the feasible solution space, we can output intermediate solutions at any point to observe the trajectory of solution quality. In our experiments, we set time slots for these improvement-based algorithms to track the variation in solution quality at different running times. Given that 2-opt operates quite swiftly in practice, its descent process is rapid and almost appears as a straight line.

## J LIMITATIONS AND BROADER IMPACTS

**Limitations.** The main limitation of this paper lies in the fact that our primary research revolves around ML4TSP and has not yet fully explored all other problems. While we aim to provide insights into the entire ML4CO domain, on the other hand, we intend to thoroughly deconstruct the existing system to explore various aspects of the current major research. Therefore, we have chosen the widely studied and researched TSP problem for comprehensive decomposition and investigation. This also means that the workload of our work solely on ML4TSP is already substantial. We believe that our exhaustive exploration of TSP could also provide valuable insights and reference points for the CO community. Moreover, the framework can be adapted to other CO problems, where the methods requiring reconstruction from scratch are primarily confined to problem-specific heuristic searches.

For the importance of studying ML4TSP and its reference value for other CO problems, we consider TSP as one of the most extensively studied CO problems with the richest learning and searching techniques, making this analysis representative for ML4CO. Note numerous papers like Fu et al. (2021); Min et al. (2023); Joshi et al. (2019); Hudson et al. (2022); da Costa et al. (2020) concentrate on TSP, and it also maintains its own surveys (Pop et al., 2023; Anbuudayasankar et al., 2014; Cheikhrouhou & Khoufi, 2021; Matai et al., 2010) and competitions like AI4TSP (Zhang et al., 2023), highlighting its significant importance. Meanwhile, any other NP problems can reduced to TSP in poly-time (since it is NP-hard) to solve, and problems like VRP can directly be broken down to TSP (Luo et al., 2024).

**Broader Impacts.** The proposed principled and comprehensive framework for tackling combinatorial optimization problems can address the lack of unification and offer guidance on crucial design principles for ML4TSP and can be generalized to ML4CO. The demonstrated advantages of specific techniques, along with the development of new solvers, are expected to inspire further research and innovation in the field. The strategic decoupling of existing methods and the unified modular framework that reproduces or even enhances the performance of existing methods can serve as the codebase and the methodology pool for further research and engineering practices. The paper's insights and observations underscore its potential to enhance the effectiveness of ML-based approaches in solving complex optimization problems, ultimately benefiting various application domains that rely on combinatorial optimization.

