# OpenReview forum: "Unify ML4TSP: Drawing Methodological Principles for TSP and Beyond from Streamlined Design Space of Learning and Search"
_ICLR.cc/2025/Conference — ICLR 2025 Poster_

### Official Review · Reviewer_Psci · 2024-10-25

**Soundness:** 3
**Presentation:** 3
**Contribution:** 3
**Rating:** 6
**Confidence:** 3

**Summary:**

In this paper, a framework that puts the most meaningful works of ML for combinatorial optimization within a unified streamline is proposed. The aim is to provide a general view of the progress being made in the machine learning field for solving combinatorial optimization problems. The exercise is illustrated by taking as a case study the Travelling Salesperson Problem (TSP). Thanks to the unified view, the authors propose a series of modifications that permit the improvement of the performance of existing approaches.

**Strengths:**

S1. Giving a general view of the works done in the area is interesting, especially if connections between the different works are made, as is the case.

S2.The paper is well written, although at times it becomes heavy to read—especially the abstract and the introduction. Later on, however, it becomes more readable.

S3. Authors have proposed improvements for existing algorithms in the different stages of the streamlining, and they resulted in effective. This is a positive aspect since the paper is not merely a review of existing algorithms. In my opinion, this has value.

**Weaknesses:**

W1. The main problem of this paper is its case study. In my opinion, the ML field has gone too far optimizing the TSP and leaving aside other combinatorial problems. It seems that if you don't do the TSP your work is worthless, and curiously, the worthless one is the TSP, since real problems that resemble the TSP are minimal. Not so the VPR (among many others) which has almost direct applications. At the beginning of the abstract, it is made explicit that the TSP is used as a concrete case study, and it is suggested that this method that has been carried out is much more general. However, throughout the paper, no mention is made of any other problem, nor in the future work. Seen in this light, this paper looks like one more iteration of a loop that has been proposing ML models for TSP for years, increasingly better, but also increasingly isolated from real-world problems.

W2. At the beginning of section 2, the authors gave the notation for CO problems, but I got confused. But I am not clear what the authors mean - do you mean that you focus on combinatorics problems that can be represented as graphs, or do you mean that all combinatorics problems can be represented as graphs? The former would make sense to me, although you should be more precise, which opens another door, and that is... What about those that are not graphs? The second of the options I am not sure, but I would say that at least naturally not all problems can be represented as graphs.

W3. It almost goes unnoticed, but it should be noted that the reference solutions used have been obtained with LKH-3, a heuristic algorithm (if I am not mistaken), so we do not know the real solutions. At this point, the best thing to do would have been to use solvers. In the case of the TSP, Concorde is capable of arriving at optimal solutions for instances with thousands of cities. Moreover, in this line, it is also obvious that the ML models being developed for CO lag behind metaheuristic algorithms in terms of performance and execution time (if we take into account the training time). I understand that it is not the purpose of the authors to pretend to obtain SOTA results in this paper, but systematically obviating a whole family of algorithms that a priori is better than ML is not a good behavior. I think that incorporating metaheuristic algorithms as a reference would help to have a clearer view of the performance of the analyzed models.

**Questions:**

Any progress in the weaknesses noted above would be positive for the paper. In addition, I have some questions related:

Q1. Being aware that at this stage it is difficult to add a new problem to the approach that is being taken... could the authors extend the work in this direction? If the authors can do this, I am willing to reconsider the score assigned.

Q2. In Section 2, the authors state that the objective function is a function $f(\cdot, G):\{0,1\}^N \rightarrow \mathbb{R}_{\geq 0}$. However, the authors have also stated that $\Omega$ denotes the feasible set of solutions, so I understand that f should be as follows: $f(\cdot, G): \Omega \rightarrow \mathbb{R}_{\geq 0}$. Otherwise, I do not think it makes sense to map unfeasible solutions with the objective function. Please address this point.

---

### Official Review · Reviewer_i2ep · 2024-11-04

**Soundness:** 3
**Presentation:** 3
**Contribution:** 3
**Rating:** 6
**Confidence:** 5

**Summary:**

This paper proposes a unified framework for learning-based approaches for the traveling salesman problem.  This paper identifies key algorithmic components of many recent ML4TSP algorithms, e.g., prediction, solution construction, and search.
Through this lens, the authors propose a modular framework and implementation that streamlines the algorithmic choices for ML4TSP solvers.  The authors perform extensive numerical comparisons.

**Strengths:**

- Overall, the paper is well-organized and easy to read.  One strength in this case is the reviewing, discussing, and modularizing of the related work and recent efforts on ML4TSP.
- One significant contribution of this paper is the potential to lead to more streamlined and reproducible implementations of ML4TSP solvers.
- Extensive numerical comparisons.  As this paper is primarily computational, the authors extensively compare various algorithmic components in ML4TSP solvers and identify a few key elements that lead to reasonable quality solutions, i.e., MCTS.

**Weaknesses:**

- **Limitation in scope.** Perhaps the most significant weakness in the paper is the limitation in the scope of TSP.  Compared to other CO problems, e.g., the vehicle routing problem, TSP has more efficient specialized solvers, i.e., Concorde.  For this reason, the practical use case for a framework so specialized to TSP may not be impactful outside of studying the trade-off with learning-based methods. As such, one way to strengthen the contribution practically would be to expand beyond only TSP to include CO problems wherein state-of-the-art methods are not as scalable as those for TSP.  The authors do mention this as a limitation of the work.   Including more difficult CO problems with more complex structures would be much appreciated.
- **Evaluation on "Easy" Instances.** Even the largest set of instances, i.e., TSP500, can be solved optimally in 18 seconds, with algorithms like LKH3 producing the same quality solutions in less than one second.  Given that the paper focuses explicitly on only TSP, evaluating more challenging instances would be helpful given the limitation in scope.
- **No Code to Review.** The authors mention that the code will be released upon submission.  However, given this paper is intended to give a modular view and implementation, the fact that no anonymized code is included in the submission makes reviewing the paper challenging.  For instance, based on the paper, it is impossible to tell how easy implementing other CO problems would be, or even other algorithmic components.   Assuming the implementation is well done, I would certainly be inclined to raise my score.

**Questions:**

**Questions**
- How difficult is extending the existing library to other CO problems, i.e., VRP, or even CO problems not on graphs?
- Are there any methods in the ML4TSP space that do not fit into the implementation/view proposed?

**Minor Remarks**
- Some of the bolding in tables is missing or even incorrect, e.g., Table 6 has two different times bolded, which should not be the case.   In the final version of the paper, I would encourage the authors to review the tables/bolding carefully.
- The length metric is not that useful.  Overall, it makes the tables harder to read.  At least in the main paper, I would encourage using a single metric to report solution quality, i.e., the gap to the best-known or reference solution.
- Some of the earlier papers on ML4TSP are not mentioned/discussed, i.e., [1,2,3]

**References**

[1] Elias Khalil, Hanjun Dai, Yuyu Zhang, Bistra Dilkina, and Le Song. Learning combinatorial optimization algorithms over graphs. Advances in neural information processing systems, 30, 2017.

[2] Alex Nowak, Soledad Villar, Afonso S Bandeira, and Joan Bruna. A note on learning algorithms for
quadratic assignment with graph neural networks. stat, 1050:22, 2017.

[3] Oriol Vinyals, Meire Fortunato, and Navdeep Jaitly. Pointer networks. Advances in neural information
processing systems, 28, 2015

---

### Official Review · Reviewer_sguF · 2024-11-04

**Soundness:** 2
**Presentation:** 2
**Contribution:** 2
**Rating:** 5
**Confidence:** 4

**Summary:**

This paper systematically explores the design space of machine learning-based approaches for the Traveling Salesman Problem (TSP) and introduces a unified framework for analyzing and enhancing these solutions. By decomposing existing methods into modular components of learning and search, the paper identifies key principles that improve solution quality, such as joint probability estimation, symmetry handling, and efficient exploration-exploitation strategies. Through a comprehensive set of ablation studies, the authors reveal the impact of each design choice and propose novel configurations that outperform current learning-based TSP solvers.

**Strengths:**

- The paper introduces a systematic framework for analyzing and enhancing ML-based TSP solvers by decomposing methods into modular components, offering new insights into the design space.

- This paper shows extensive experimental study, thoroughly examining various component and dataset combinations and offering a comprehensive evaluation of the approach.

- The implementation is well-structured, supporting straightforward development and extension for future research and engineering applications.

- The paper is well-organized and clearly presents its findings.

**Weaknesses:**

- A primary concern is that the study’s conclusions are derived solely from experiments on the TSP, without sufficient exploration of their applicability to other routing problems or broader CO challenges. While TSP is a classic problem and valuable as a benchmark, part of its relevance comes from its potential to inform solutions for a variety of routing and CO problems via machine learning approaches. Given the extensive experiments conducted, the paper would be significantly strengthened by discussing how the findings and insights might generalize beyond TSP, suggesting implications or adaptations for other problem types. Without this, readers may question the broader utility of the proposed framework.

- Another potential weakness is the study’s strict comparison of components based solely on optimality, without considering varying objectives that real-world applications may require. In some scenarios, the primary goal may be to find an optimal solution when sufficient time is available, while in other cases, the focus may be on obtaining a good approximate solution within a limited time frame. By focusing only on optimality, the comparisons risk being somewhat misleading, as certain components may be preferable in scenarios that prioritize speed or approximation quality over strict optimality. Acknowledging these different practical objectives and testing components under varying solution scenarios would give a more balanced and realistic view of each component’s strengths.

**Questions:**

In Tables 7 and 8, the results show that the best-performing configuration varies depending on the component and dataset combination. Could the authors clarify how these variations should be interpreted? Specifically, does this indicate that no single model consistently performs best across all scenarios, or are there insights into why certain components perform better with particular datasets? A clearer explanation here would help determine whether the findings suggest that different settings require tailored configurations rather than a universally optimal model.

---

### Official Review · Reviewer_7Hhi · 2024-11-05

**Soundness:** 3
**Presentation:** 3
**Contribution:** 3
**Rating:** 8
**Confidence:** 3

**Summary:**

This paper explores machine learning-based methods for solving the Traveling Salesman Problem (TSP) and proposes a modular framework to streamline the design space of ML4TSP solvers. By systematically decomposing established learning-based TSP solvers, the authors develop a unified, modular approach that combines learning and search functionalities. This framework facilitates ablation studies, making it easier to assess the specific contributions of learning components and search strategies in current approaches. Key principles identified include joint probability estimation, symmetry solution representation, and online optimization, which enhance the efficiency and quality of ML4TSP solutions. The paper also proposes new combinations and enhancements to existing methods, improving performance and providing insights into optimal configurations for future ML-based combinatorial optimization research.

**Strengths:**

Originality: The paper presents a novel approach by developing a modular framework to dissect and streamline the design space of ML-based TSP solvers, which is a fresh and valuable perspective.

Quality: The research is thorough, with a clear methodology that includes well-defined design principles, such as joint probability estimation, symmetry solution representation, and online optimization. The paper carefully analyzes these principles through comprehensive ablation studies, making the results robust and reliable.

Clarity: The paper is well-structured, with a clear and logical progression from problem background to framework design, implementation, and results. Each section explains the functionality and impact of various design components within the modular framework, making it easy for readers to follow.

Significance: The paper addresses a crucial gap in the field of ML for combinatorial optimization by providing a structured approach to analyze and optimize TSP solvers.

**Weaknesses:**

Lack of Theoretical Justification for Design Principles: While the paper empirically demonstrates the effectiveness of the proposed design principles (e.g., joint probability estimation and symmetry solution representation), it lacks a theoretical foundation to support it.

Limited Exploration of Scalability: Although the framework is tested on TSP instances of varying sizes, there is limited discussion on its scalability to very large TSP instances or other complex combinatorial problems.

Limited Generalization to Other Optimization Problems: Although the framework is designed with TSP in mind, the paper could benefit from explicitly discussing its adaptability to other combinatorial optimization problems (e.g., Vehicle Routing, Knapsack, or Job Scheduling).

**Questions:**

Could the authors expand the comparison with traditional, non-ML solvers?

---

### Meta-Review · Area_Chair_QCMX · 2024-12-22

**Metareview:**

This paper proposed a unified modular streamline for machine learning-based approaches for the Traveling Salesman Problem (TSP). It decomposes existing methods into modular components of learning and search, and proposed enhancements to existing method based on empirical findings. Reviewers found this work interesting and well-written, as well as its extensive evaluation with valuable findings. In terms of weaknesses, a common limitation pointed out by the reviewers is that this work focuses only on TSP. This issue has been addressed by the authors during rebuttal. They show that the proposed framework can generalize to other problems. I recommend acceptance, and urge authors to have a more detailed discussion and empirical evaluation on other representative CO problems, and open source their implementation for facilitate future research in the NCO community.

**Additional Comments On Reviewer Discussion:**

Authors did a good job in addressing all reviewers' comments. Additional results and explantions are provided, and three reviewers raised their score.

---

### Decision · Program_Chairs · 2025-01-22

Accept (Poster)